# Alcohol Consumption and Liver Metabolism in the Era of MASLD: Integrating Nutritional and Pathophysiological Insights

**DOI:** 10.3390/nu17132229

**Published:** 2025-07-05

**Authors:** Carlo Acierno, Fannia Barletta, Alfredo Caturano, Riccardo Nevola, Ferdinando Carlo Sasso, Luigi Elio Adinolfi, Luca Rinaldi

**Affiliations:** 1Department of Infectious Diseases, San Carlo Hospital, 85100 Potenza, Italy; 2Department of Anesthesiology and Intensive Care, San Carlo Hospital, 85100 Potenza, Italy; fannia.barletta@ospedalesancarlo.it; 3Department of Human Sciences and Promotion of the Quality of Life, San Raffaele Roma Open University, 00166 Rome, Italy; 4Liver Unit, “A. Landolfi” Hospital, AORN S. G. Moscati, 83029 Solofra, Italy; riccardo.nevola@unicampania.it; 5Department of Advanced Medical and Surgical Sciences, University of Campania “Luigi Vanvitelli”, 80138 Napoli, Italy; ferdinandocarlo.sasso@unicampania.it (F.C.S.); luigielio.adinolfi@unicampania.it (L.E.A.); 6Department of Medicine and Health Science “Vincenzo Tiberio”, Università Degli Studi del Molise, 86100 Campobasso, Italy; luca.rinaldi@unimol.it

**Keywords:** MASLD, MetALD, CYP2E1, mitochondrial dysfunction, oxidative stress, sarcopenia, micronutrient deficiencies, nutritional counseling in liver disease

## Abstract

Metabolic dysfunction-associated steatotic liver disease (MASLD) has emerged as the leading cause of chronic liver disease worldwide, driven by the global epidemics of obesity, type 2 diabetes, and metabolic syndrome. In this evolving nosological landscape, alcohol consumption—traditionally excluded from the diagnostic criteria of non-alcoholic fatty liver disease (NAFLD)—has regained central clinical importance. The recently defined MetALD phenotype acknowledges the co-existence of metabolic dysfunction and a significant alcohol intake, highlighting the synergistic nature of their pathogenic interactions. This narrative review provides a comprehensive analysis of the biochemical, mitochondrial, immunometabolic, and nutritional mechanisms through which alcohol exacerbates liver injury in MASLD. Central to this interaction is cytochrome P450 2E1 (CYP2E1), whose induction by both ethanol and insulin resistance enhances oxidative stress, lipid peroxidation, and fibrogenesis. Alcohol also promotes mitochondrial dysfunction, intestinal barrier disruption, and micronutrient depletion, thereby aggravating metabolic and inflammatory derangements. Furthermore, alcohol contributes to sarcopenia and insulin resistance, establishing a bidirectional link between hepatic and muscular impairment. While some observational studies have suggested a cardiometabolic benefit of a moderate alcohol intake, emerging evidence challenges the safety of any threshold in patients with MASLD. Accordingly, current international guidelines recommend alcohol restriction or abstinence in all individuals with steatotic liver disease and metabolic risk. The review concludes by proposing an integrative clinical model and a visual cascade framework for the assessment and management of alcohol consumption in MASLD, integrating counseling, non-invasive fibrosis screening, and personalized lifestyle interventions. Future research should aim to define safe thresholds, validate MetALD-specific biomarkers, and explore the efficacy of multidisciplinary interventions targeting both metabolic and alcohol-related liver injury.

## 1. Introduction

Over the past two decades, chronic liver diseases have undergone a profound epidemiological transformation, paralleling the global rise in the incidence of metabolic syndrome and its complications [1]. In this context, the recent introduction of the term metabolic dysfunction-associated steatotic liver disease (MASLD) represents a paradigm shift in the classification of liver disorders, redirecting diagnostic focus from the mere exclusion of alcohol consumption—as required by the former definition of NAFLD—to the active identification of metabolic dysfunction as a key diagnostic criterion [2].

MASLD is now recognized as the most prevalent form of chronic liver disease, with an estimated prevalence ranging from 30% to 40% in the general adult population, and substantially higher rates among individuals with obesity, type 2 diabetes, or dyslipidemia [3,4]. According to a recent agent-based simulation study, the prevalence of MASLD in the United States is projected to rise from 33.7% in 2020 to 41.4% by 2050, with a fourfold increase in liver transplants due to metabolic causes anticipated over the same period [5]. These findings underscore the urgent need for a comprehensive understanding of the disease’s pathogenic determinants.

The new nosological framework has also introduced the category of MetALD (Metabolic and Alcohol-related Liver Disease), which identifies individuals with hepatic steatosis and metabolic dysfunction who consume alcohol in amounts above conventional thresholds (≥20 g/day for women, ≥30 g/day for men) [6]. This classification transcends the traditional dichotomy between metabolic and toxic etiologies, acknowledging the synergistic nature of the interactions between metabolic dysfunction and alcohol exposure [7]. In this framework, alcohol is no longer considered an exclusion criterion but rather a pathogenic cofactor with significant diagnostic, prognostic, and therapeutic implications.

A growing body of evidence suggests that even a “moderate” alcohol intake may exacerbate mitochondrial oxidative stress, amplify hepatic inflammation, and promote fibrogenesis in patients with MASLD, thereby facilitating progression to steatohepatitis (MASH), cirrhosis, or hepatocellular carcinoma [8]. These effects have been documented in both preclinical models and prospective cohorts, revealing a marked increase in liver-related mortality among individuals with MASLD who consume excessive amounts of alcohol, compared to those affected by either condition alone [9,10,11]. Moreover, recent meta-analyses challenge the purported “safety” of moderate alcohol consumption in the setting of metabolic dysfunction, highlighting the absence of a universally protective threshold [12]. In light of this evidence, alcohol intake—even in the absence of overt abuse—assumes a central role in the natural history of MASLD and should be integrated into risk stratification strategies.

In this narrative review, we aim to critically and comprehensively examine the principal pathogenic mechanisms—biochemical, mitochondrial, immunometabolic, and nutritional—by which alcohol exacerbates liver injury in the setting of MASLD. Specific attention will be devoted to CYP2E1 induction, mitochondrial dysfunction, the impairment of the gut barrier, micronutrient depletion, and muscular complications, particularly sarcopenia. Finally, we will discuss the clinical and nutritional implications, the concept of alcohol “moderation,” and potential personalized counseling strategies. The objective is to provide an up-to-date, clinically applicable overview, with evidence-based, multidisciplinary management proposals.

Table 1 outlines the clinical, diagnostic, and pathophysiological distinctions among MASLD, MetALD, MASH, and ALD, highlighting their overlapping features and unique determinants.

### Updated Nomenclature and Definitions of Steatotic Liver Disease

In 2023, an international expert consensus replaced the term “non-alcoholic fatty liver disease” (NAFLD) with metabolic dysfunction-associated steatotic liver disease (MASLD), in recognition of the central role of metabolic dysregulation in the pathogenesis of the condition [14].

This reclassification seeks to enhance diagnostic clarity and reduce stigma by defining the disease according to its underlying causative mechanisms rather than by exclusion criteria:MASLD is diagnosed in the presence of hepatic steatosis (≥5% liver fat as determined by imaging or histology) and at least one of the following five cardiometabolic risk factors:○Overweight or obesity (BMI ≥ 25 kg/m^2^ or ethnicity-specific thresholds);○Type 2 diabetes or prediabetes;○Arterial hypertension (≥130/85 mmHg or current antihypertensive treatment);○Hypertriglyceridemia (≥150 mg/dL) or treatment with lipid-lowering agents;○Low HDL-cholesterol (≤40 mg/dL in men, ≤50 mg/dL in women) or lipid-lowering therapy [3].MetALD (Metabolic and Alcohol-related Liver Disease) refers to individuals with hepatic steatosis and metabolic dysfunction who consume alcohol in amounts above established threshold levels (typically > 20 g/day for women and >30 g/day for men). This category acknowledges the clinical overlap between metabolic and alcohol-induced liver injury [4].The term MASH (metabolic dysfunction-associated steatohepatitis) replaces “NASH” and designates steatohepatitis, with or without fibrosis, arising in the context of MASLD [3].

This revised taxonomy facilitates the classification of patients with mixed etiologies (e.g., MASLD coexisting with hazardous alcohol consumption) and promotes a more inclusive, pathophysiologically grounded framework for the categorization of fatty liver diseases.

## 2. Hepatic Alcohol Metabolism and Metabolic Adaptations

### 2.1. Biochemical Pathways of Ethanol Oxidation

Ethanol metabolism occurs predominantly in the liver via three major pathways: oxidative, microsomal, and non-oxidative routes [15]. The oxidative pathway is the primary mechanism under physiological conditions and is catalyzed by alcohol dehydrogenase (ADH) in the cytoplasm, which converts ethanol to acetaldehyde, a highly reactive and toxic intermediate [16]. Acetaldehyde is subsequently oxidized to acetate by mitochondrial aldehyde dehydrogenase 2 (ALDH2), a process that concurrently generates NADH, leading to a significant alteration in the mitochondrial NAD^+^/NADH ratio [15]. This redox imbalance impairs hepatic fatty acid oxidation and inhibits gluconeogenesis, thereby contributing to triglyceride accumulation and hepatic steatosis (Arumugam MK, 2023) [17]. Concurrently, chronic alcohol metabolism activates the microsomal ethanol oxidizing system (MEOS), primarily mediated by cytochrome P450 2E1 (CYP2E1), which is localized in the smooth endoplasmic reticulum and also in hepatic mitochondria [15]. This inducible enzyme, upregulated by both ethanol and insulin resistance, generates reactive oxygen species (ROS) during ethanol oxidation, promoting lipid peroxidation, mitochondrial injury, and oxidative stress [18]. CYP2E1 hyperactivity is exacerbated in individuals with MASLD, rendering this population particularly vulnerable to the hepatotoxic effects of alcohol [19]. A smaller fraction of ethanol is metabolized via non-oxidative pathways, resulting in the formation of fatty acid ethyl esters (FAEEs), phosphatidylethanol (PEth), ethyl sulfate (EtS), and ethyl glucuronide (EtG) [20]. These metabolites, catalyzed by FAEE synthases, phospholipase D, and UDP-glucuronosyltransferases, serve not only as sensitive biomarkers of alcohol consumption but they also exert direct cytotoxic effects on mitochondrial and cellular membranes [21].

The principal routes of ethanol metabolism—both oxidative and non-oxidative—play a critical role in shaping the hepatic redox environment and promoting lipotoxicity in MASLD. The stepwise conversion of ethanol to acetaldehyde and subsequently to acetate, along with the generation of NADH and reactive oxygen species (ROS), underscores the biochemical basis of alcohol-induced hepatic injury (Figure 1).

### 2.2. Metabolic Alterations Induced by Chronic Alcohol Exposure

The excess NADH generated by ethanol metabolism profoundly alters the hepatic metabolic profile: it inhibits mitochondrial β-oxidation (via the accumulation of NADH and malonyl-CoA), activates de novo lipogenesis (through ACC, SREBP-1c, and ChREBP), suppresses gluconeogenesis (by shifting pyruvate toward lactate), and impairs oxidative phosphorylation by disrupting the Krebs cycle and electron transport chain [22,23]. These disturbances promote intrahepatic triglyceride accumulation, further exacerbated by the activation of the lipogenic transcription factor SREBP-1c and the repression of PPARα [24]. The accumulation of acetaldehyde, a reactive intermediate of ethanol metabolism, significantly compromises the efficiency of mitochondrial redox shuttles, particularly the malate–aspartate and α-glycerophosphate shuttles [25]. This exacerbates redox imbalance by impairing cytosolic NAD^+^ regeneration, thereby worsening the inhibition of oxidative phosphorylation [25,26,27]. CYP2E1-induced overproduction of ROS damages components of the respiratory chain—especially complexes I and III—causing mitochondrial membrane depolarization, cytochrome *c* release, and the activation of the intrinsic apoptotic pathway in hepatocytes [28]. In animal models and human hepatocytes chronically exposed to ethanol, observations include megamitochondria, disrupted mitochondrial fusion and fission dynamics, and impaired mitophagy, all of which contribute to persistent bioenergetic dysfunction [29]. Moreover, mitochondrial glutathione (mtGSH)—which is essential for ROS detoxification—is depleted due both to increased consumption and to reduced mitochondrial membrane permeability [30]. An early ethanol-induced phenomenon is the “swift increase in alcohol metabolism” (SIAM), characterized by increased mitochondrial oxygen consumption and NADH turnover, leading to inefficient ATP production [31].

These adaptive responses culminate in a persistent state of oxidative stress, metabolic instability, and heightened hepatocellular vulnerability.

### 2.3. Distinct and Shared Mechanisms of Hepatic Injury in Alcoholic vs. Metabolic Dysfunction

Although alcohol-related liver injury and liver dysfunction associated with metabolic impairment arise from distinct etiological factors, they share common pathogenic pathways with significant clinical implications. Ethanol metabolism results in a pronounced excess of NADH and accumulation of acetaldehyde, inducing profound redox imbalance, lipotoxicity, and mitochondrial toxicity [32]. Furthermore, only alcohol metabolism leads to the formation of non-oxidative ethyl metabolites such as fatty acid ethyl esters (FAEEs) and phosphatidylethanol (PEth), which are absent in patients with MASLD alone [33]. In contrast, in MASLD, hepatic dysfunction is primarily driven by chronic nutritional excess, hyperinsulinemia, and adipose tissue–derived systemic inflammation, which collectively promote insulin resistance, steatosis, and hepatic inflammation [34,35].

Nevertheless, both conditions converge on shared molecular mechanisms, including mitochondrial injury, ROS overproduction, lipid peroxidation, and the inhibition of β-oxidation [36,37].

This shared metabolic–oxidative axis is particularly relevant in individuals with dual exposure (MetALD), who exhibit synergistic CYP2E1 expression, heightened inflammation, increased oxidative stress, and accelerated progression toward MASH, advanced fibrosis, and cirrhosis [38,39].

## 3. Mitochondrial Dysfunction and Oxidative Stress: The Converging Axis of Alcohol and MASLD

### 3.1. Impact on Mitochondrial Bioenergetics and ROS Generation

Hepatocellular mitochondria are the primary targets of ethanol-induced toxicity, as they are central to fatty acid β-oxidation and energy production [40].

Alcohol metabolism—particularly through the induction of CYP2E1—leads to an excess of reducing equivalents (NADH) and a direct increase in the generation of reactive oxygen species (ROS), disrupting the NAD^+^/NADH ratio and impairing the function of the mitochondrial electron transport chain (ETC) [41]. Chronic accumulation of mitochondrial ROS compromises oxidative phosphorylation (OXPHOS), and reduces ATP synthesis, induces the transient hyperpolarization of the mitochondrial membrane potential (ΔΨm), followed by its collapse, triggering cytochrome *c* release and the activation of the intrinsic apoptotic pathway [42].

Studies in patients with alcoholic steatosis or MASLD have documented a significant reduction in the activity of respiratory chain complexes I and III, accompanied by the elevated production of hydrogen peroxide (H_2_O_2_) and superoxide anion (O_2_^−^), resulting in oxidative damage to proteins, lipids, and DNA [43]. These processes are further exacerbated by mitochondrial glutathione (mtGSH) depletion—due to both increased consumption and reduced membrane permeability—and by the mitochondrial overexpression of CYP2E1, which amplifies ROS production within the intermembrane space [44].

The interplay between oxidative stress, respiratory dysfunction, and apoptosis establishes a vicious bioenergetic cycle that perpetuates hepatocellular injury, particularly in individuals with concomitant metabolic dysfunction [45].

In MASLD, similar mechanisms are triggered by excess nutrients (glucose, fatty acids, fructose), which induce mitochondrial lipotoxicity and respiratory uncoupling [46]. Moreover, patients with MASH exhibit persistent markers of oxidative damage, such as 8-hydroxy-2′-deoxyguanosine (8-OHdG) and malondialdehyde (MDA), reflecting an inadequate antioxidant capacity and serving as predictors of fibrotic progression [47].

The biochemical and structural impacts of alcohol metabolism on hepatic mitochondria in MASLD are detailed in Table 2.

This diagram complements Table 2 by illustrating the mitochondrial cascade triggered by oxidative stress in alcohol-exposed MASLD. The progressive sequence—from CYP2E1 induction and ROS overproduction to ATP depletion and apoptosis—highlights the bioenergetic failure underlying hepatocellular damage in this condition (Figure 2).

### 3.2. Lipid Homeostasis Disruption and Hepatocellular Injury

Mitochondrial oxidative stress profoundly disrupts hepatic lipid homeostasis, promoting the accumulation of lipotoxic species and triggering inflammatory and fibrogenic signaling cascades [34,37]. Inhibition of mitochondrial β-oxidation—resulting from ETC complex damage and the dysfunction of carnitine palmitoyltransferase I (CPT1)—leads to the intracellular accumulation of free fatty acids (FFAs), acyl-CoA, diacylglycerols (DAGs), and ceramides. These bioactive lipids possess pro-apoptotic and pro-inflammatory properties [48]. Such toxic lipid intermediates activate protein kinases such as PKC, inflammatory pathways including JNK/NF-κB, and toll-like receptors (TLRs), thereby amplifying sterile immune responses [36]. In particular, DAGs and ceramides promote endoplasmic reticulum (ER) stress and further mitochondrial ROS generation, reinforcing a dysfunctional redox environment [49]. Secondary lipid aldehydes such as 4-hydroxynonenal (4-HNE) and malondialdehyde (MDA) form protein and DNA adducts, stimulating the activation of Kupffer cells and hepatic stellate cells (HSCs) [50]. Multiple clinical studies have shown that individuals with the MetALD phenotype—defined by the coexistence of metabolic dysfunction and alcohol consumption—exhibit significantly higher levels of transaminases, fibrotic markers (e.g., PIIINP, TIMP-1), and oxidative stress biomarkers than those with MASLD alone, suggesting a synergistic effect of alcohol on fibrotic progression [38,51]. Moreover, CYP2E1 overexpression in MetALD models further enhances lipid peroxidation and ROS production, aggravating hepatocellular inflammation and perisinusoidal collagen deposition [52].

This molecular convergence of lipid accumulation, oxidative stress, and chronic inflammation facilitates the transition from steatosis to steatohepatitis and increases the risk of advanced fibrosis, portal hypertension, and hepatocellular carcinoma [53].

## 4. Gut–Liver Axis and Intestinal Permeability in MASLD and Alcohol-Related Liver Disease

### 4.1. Pathophysiological Role of the Gut–Liver Axis

The intestine plays a pivotal role in the pathophysiology of both metabolic and alcohol-related liver diseases through a functional interplay known as the “gut–liver axis” [54]. This bidirectional interaction is mediated by the portal circulation, the gut microbiota, and the mucosal immune system, which collectively regulate the balance between hepatic tolerance and inflammation [55]. In the setting of metabolic dysfunction and chronic alcohol consumption, the gut–liver axis becomes particularly vulnerable, fostering the activation of synergistic pathogenic mechanisms [56]. Numerous studies have shown that chronic alcohol intake profoundly alters the composition of the gut microbiota, promoting a dysbiotic state characterized by the depletion of short-chain fatty acid (SCFA)-producing bacteria—such as *Faecalibacterium prausnitzii*—and the enrichment of potentially pathogenic species such as *Escherichia coli* and *Klebsiella pneumoniae* [57]. Similarly, MASLD is associated with reduced microbial diversity, the loss of regulatory metabolic functions, and chronic mucosal inflammation [58].

The coexistence of alcohol exposure and metabolic dysfunction amplifies these alterations, intensifying the pathogenic impact of dysbiosis on liver function and leading to the profound disruption of intestinal homeostasis [59].

### 4.2. Intestinal Barrier Dysfunction and Bacterial Translocation

One of the principal consequences of dysbiosis is the disruption of the intestinal epithelial barrier, characterized by the reduced expression of tight junction proteins—such as ZO-1 and occludin—and the activation of mucosal inflammation [60]. In this context, alcohol increases intestinal permeability, facilitating the translocation of bacterial endotoxins—such as lipopolysaccharide (LPS)—and other pathogen-associated molecular patterns (PAMPs) into the portal circulation [60]. The interaction of LPS with Toll-like receptor 4 (TLR4) on Kupffer cells triggers the production of pro-inflammatory cytokines (TNF-α, IL-1β, IL-6), activating a sterile immune response that contributes to disease progression from steatosis to steatohepatitis [61].

This process is schematically illustrated in Figure 3, which depicts how intestinal dysbiosis and LPS translocation promote hepatic inflammation via Kupffer cell activation in alcohol-exposed MASLD.

Moreover, murine models have demonstrated that the pharmacologic or nutritional restoration of intestinal permeability significantly reduces alcohol-induced liver injury, confirming the causal role of microbial translocation [62,63]. In addition to LPS, microbial metabolites such as deoxycholic acid (DCA), ethylphenol, and microbially derived acetaldehyde also contribute to hepatic dysfunction through direct toxic effects and by activating hepatic nuclear receptors such as FXR and PXR [64].

These metabolites further modulate hepatic gene expression, exacerbating lipogenesis and inflammation [65].

Table 3 summarizes the alterations in intestinal integrity and gut–liver communication in alcohol-exposed MASLD.

### 4.3. Clinical Implications and Therapeutic Opportunities

Disruption of the intestinal barrier and dysbiosis contribute to the chronicity of liver injury in patients with MASLD, particularly in the context of alcohol consumption [66]. This condition increases the propensity for progression to MASH, fibrosis, and systemic complications [58]. Integrating the assessment of intestinal barrier function into the clinical management of patients with MASLD—especially those with chronic alcohol consumption—should be strongly recommended. Non-invasive biomarkers such as I-FABP and LPS, along with emerging mucosal indicators (defensins 5/6, L-FABP) and functional permeability tests (lactulose–mannitol), provide a multiparametric approach for the early detection of intestinal dysfunction [67,68,69]. Incorporating these parameters into prognostic models (e.g., FIB-4, ELF, multifactorial scores) may enhance the sensitivity and specificity of hepatic risk stratification, enabling more targeted and timely interventions. Therapeutic modulation of the gut microbiota—including the use of probiotics, prebiotics, postbiotics, and selective TLR antagonists (e.g., TLR4 inhibitors)—represents an emerging strategy to target the gut–liver axis, particularly in the MetALD phenotype [70,71]. These approaches improve intestinal barrier integrity, reduce endotoxemia, and inhibit hepatic inflammatory pathways (via TLR4/NF-κB), thereby attenuating disease progression [72].

## 5. Pathophysiological Interaction Between Alcohol and MASLD

### 5.1. Shared Mechanisms and Synergistic Hepatotoxicity

The interaction between alcohol consumption and metabolic steatotic liver disease (MASLD) represents not merely an additive risk profile, but a genuine pathogenetic synergism.

This synergistic interaction is conceptually illustrated in Figure 4, where overlapping pathways such as oxidative stress, TLR4-mediated inflammation, and fibrogenesis are shown to arise from both alcohol-induced and metabolic insults.

Both conditions share overlapping metabolic and molecular pathways that can mutually exacerbate hepatocellular injury [56]. A key point of convergence is cytochrome P450 2E1 (CYP2E1), whose expression is induced by both chronic ethanol intake and the pro-inflammatory, insulin-resistant milieu that is typical of MASLD. This results in the excessive production of reactive oxygen species (ROS), promoting lipid peroxidation, mitochondrial injury, and the activation of the JNK–p38–ASK1 stress signaling cascade [73]. CYP2E1 overexpression also facilitates the bioactivation of environmental hepatotoxins (e.g., carbon tetrachloride, acetaminophen, thioacetamide), which—particularly in murine models—induce acute hepatotoxicity and fibrosis via oxidative stress and mitochondrial post-translational modifications (PTMs) [74]. In CYP2E1-knockout animals, exposure to these compounds produces markedly attenuated toxicity, underscoring the enzyme’s central role in mediating mitochondrial injury in both alcohol-related and metabolic liver disease contexts [15]. Moreover, both alcohol and MASLD disrupt intestinal barrier integrity, leading to endotoxemia and bacterial translocation—key drivers of hepatic inflammation through Toll-like receptor 4 (TLR4) activation and transformation of hepatic stellate cells into myofibroblastic phenotypes [66]. Endotoxin, in particular, functions as a “second hit” with potent pro-inflammatory and pro-fibrogenic effects in individuals with underlying hepatic steatosis [75]. Finally, concomitant alcohol use and exposure to hepatotoxic drugs—such as acetaminophen—markedly elevate the risk of fulminant hepatic necrosis, especially in individuals with glutathione depletion and pre-existing MASLD, as demonstrated in both experimental and clinical settings [76].

Figure 5 illustrates the overlapping molecular mechanisms and clinical outcomes that arise from alcohol consumption and metabolic dysfunction, reinforcing the rationale for classifying MetALD as a synergistic disease entity.

### 5.2. From Steatosis to Inflammation and Fibrosis: A Convergent Model

The pathological progression from simple steatosis to steatohepatitis and fibrosis is a multifactorial process governed by complex molecular mechanisms that are largely shared between MASLD and alcoholic liver disease (ALD) [77]. Despite differing etiologies, both conditions involve key events that drive hepatic disease progression, thereby supporting a convergent model [78]. The initial phase is characterized by the excessive intrahepatic accumulation of triglycerides and free fatty acids, arising from an imbalance between lipid synthesis, import, and oxidation [79]. In ALD, this imbalance results from inhibition of mitochondrial β-oxidation and the induction of lipogenesis mediated by elevated NADH levels. In MASLD, the driving factors include insulin resistance, hyperinsulinemia, and the activation of SREBP-1c and ChREBP [80].

The second stage is marked by lipotoxicity: the accumulation of diacylglycerols (DAGs), ceramides, and acyl-CoAs activates pro-inflammatory signaling cascades (PKC, JNK, NF-κB), promoting cytokine release (TNF-α, IL-6, IL-1β) and the recruitment of immune cells including monocytes, T lymphocytes, and neutrophils [81]. Simultaneously, CYP2E1 induction—which is predominant in ALD—and nutrient overload in MASLD amplify ROS production and redox imbalance, perpetuating oxidative stress [82]. Mitochondrial dysfunction and endoplasmic reticulum (ER) stress further drive apoptosis and necroinflammation, accompanied by the release of damage-associated molecular patterns (DAMPs), which activate Kupffer cells and hepatic stellate cells (HSCs) [28]. Upon stimulation by profibrogenic mediators such as TGF-β and PDGF, HSCs differentiate into collagen-producing myofibroblasts, initiating fibrogenesis [83]. Emerging evidence suggests that in “mixed” phenotypes such as MetALD, the coexistence of metabolic derangements and alcohol consumption exerts a synergistic effect, accelerating the transition to MASH or ASH with significant fibrosis, and increasing the risk of hepatocellular carcinoma [84].

Given these synergistic mechanisms, there is an urgent need for clinical tools that are capable of the early identification of individuals who are at a heightened risk of progression to steatohepatitis and advanced fibrosis—particularly in the context of even moderate alcohol consumption.

Table 4 compares the molecular mechanisms of MASLD and ALD, emphasizing the synergistic pathogenicity observed in MetALD.

## 6. Micronutrient Depletion and Malabsorption Syndromes

### 6.1. Deficiencies in Folate, Thiamine, Zinc, Magnesium, and B Vitamins

Chronic alcohol consumption is a major etiological factor in both subclinical and overt malnutrition among patients with liver disease, even in the absence of overt alcoholism [13]. In individuals with MASLD, the coexistence of metabolic dysfunction and ethanol intake exacerbates nutritional deficits, leading to the depletion of essential micronutrients with significant consequences for hepatic, immune, and muscular function [85].

The principal micronutrients affected include the following:Thiamine (vitamin B1) is a cofactor for pyruvate oxidative decarboxylation and the pentose phosphate pathway. Thiamine deficiency impairs oxidative glucose metabolism, promoting lactate accumulation and contributing to hepatic and cerebral dysfunction [86]. In patients with MASLD, it may worsen insulin resistance and promote progression to steatohepatitis [87].Folate is involved in the synthesis of S-adenosylmethionine (SAM) and DNA methylation. Folate deficiency induces genomic instability, reduces nucleotide synthesis, and increases hepatic oncogenic risk [88,89,90].Zinc is essential for the activity of numerous hepatic metalloenzymes, innate immune function (NK cells and neutrophils), redox homeostasis, and ammonia detoxification [91]. Zinc deficiency is associated with hepatic encephalopathy, insulin resistance, and impaired intestinal barrier integrity [92].Magnesium plays a role in mitochondrial membrane stabilization, oxidative phosphorylation, and glucose metabolism [93]. Hypomagnesemia—which is common in alcohol users—is associated with muscle cramps, hypocalcemia, and insulin resistance [94].B-complex vitamins (B2, B6, B12) are essential cofactors in multiple metabolic pathways. Their deficiency manifests as cognitive impairment, peripheral neuropathy, and disruptions in energy metabolism [95,96].

Figure 6 summarizes the main micronutrients affected by alcohol exposure in MASLD, along with their mechanisms of depletion and associated clinical consequences.

These deficiencies arise through multiple mechanisms: reduced dietary intake due to substitution with alcohol-derived “empty calories”, gut dysbiosis and intestinal inflammation, the inhibition of apical transporters (e.g., SLC19A2 for thiamine) [97], increased urinary excretion, and decreased systemic bioavailability due to the hepatic prioritization of ethanol metabolism [95,98].

Micronutrient deficiencies and their systemic effects in alcohol-exposed MASLD are detailed in Table 5.

### 6.2. Nutritional Interactions of Alcohol and Clinical Implications

Alcohol profoundly alters the absorption, metabolism, and systemic utilization of micronutrients, disrupting both intestinal mucosal integrity and hepatic metabolic function [95,98]. At the intestinal level, ethanol induces villous atrophy, reduces the expression of tight junction proteins (ZO-1, occludin), and increases paracellular permeability, thereby facilitating microbial translocation and exacerbating intestinal inflammation [99]. Ethanol exerts a rapid and direct inhibitory effect on sodium-glucose co-transport (SGLT1) and the enterocytic Na^+^/K^+^-ATPase, thereby impairing glucose absorption [100]. It also inhibits the neutral amino acid transporter B^0^AT1 (SLC6A19), reducing both protein expression and V_max_. These effects, combined with ion gradient disruption, compromise the absorption of amino acids, glucose, and likely water-soluble vitamins, significantly contributing to malnutrition in chronic alcohol consumers [95,98,99]. In hospitalized patients or those with advanced steatohepatitis and cirrhosis, the risk of micronutrient deficiency is markedly elevated due to increased metabolic demand, chronic inflammation, and reduced dietary intake [101]. In such settings, the availability of reliable serological markers is often limited or delayed. Therefore, current evidence and clinical guidelines support the early empirical supplementation of key nutrients—including water-soluble vitamins (B_1_, B_6_, folate), zinc, and fat-soluble vitamins—even in the absence of definitive laboratory confirmation [102].

Metabolically, alcohol acts as a preferential substrate, displacing other nutrients from hepatic metabolism: it diverts NAD^+^ and acetyl-CoA from cellular reactions, impairs mitochondrial lipid oxidation, gluconeogenesis, and ATP synthesis, and inhibits the activation of fat-soluble vitamins such as A and D [103]. This metabolic competition extends to transport proteins and binding molecules, further exacerbating systemic deficiencies of essential cofactors [104]. Given these mechanisms, early recognition and correction of micronutrient deficiencies in patients with MASLD and alcohol consumption constitute a clinically relevant strategy. Recent studies suggest that targeted supplementation of selected vitamins and trace elements may improve redox status, mitochondrial function, and immune response, with potential prognostic benefits [105,106].

## 7. Alcohol, Insulin Resistance, and Sarcopenia

### 7.1. Liver–Muscle Axis in MASLD

Skeletal muscle is a metabolically active organ that is functionally interconnected with the liver through a bidirectional axis that regulates energy homeostasis, amino acid metabolism, and insulin sensitivity [107]. In MASLD, hepatic dysfunction is frequently accompanied by reductions in muscle mass and strength, constituting a sarcopenic phenotype with significant prognostic implications [108]. Epidemiological studies report a prevalence of sarcopenia ranging from 17% to 40% among individuals with MASLD, with higher rates observed in patients with visceral obesity, type 2 diabetes, or advanced steatohepatitis [108,109]. The presence of sarcopenia is associated with increased risk of hepatic fibrosis, poorer glycemic control, cardiovascular dysfunction, and higher all-cause mortality [110].

The combined phenotype of MASLD and sarcopenia defines a high-risk population characterized by a reduced functional capacity, diminished responsiveness to nutritional interventions, and heightened susceptibility to infections [110].

### 7.2. Synergistic Effects of Alcohol on Muscle Mass

Alcohol acts as an additional aggravating factor for sarcopenia, disrupting anabolic signaling and impairing mitochondrial function in skeletal muscle [111]. At the molecular level, ethanol inhibits muscle protein synthesis by downregulating the IGF-1/PI3K/AKT/mTOR pathway, activating the ubiquitin–proteasome system, upregulating myostatin expression, and stimulating proteolysis [112]. Concurrently, alcohol impairs mitochondrial biogenesis in myocytes by inhibiting the expression of PGC-1α and TFAM, leading to increased ROS production and the loss of oxidative capacity [113]. The combined effect of mitochondrial dysfunction and oxidative stress contributes to muscle mass loss and reduced contractile function.

Micronutrient deficiencies—particularly in vitamin D, zinc, and magnesium—commonly observed in patients with metabolic–alcoholic liver disease, further exacerbate this pathogenic process. Specifically, vitamin D deficiency is associated with the reduced expression of nuclear receptors in muscle cells, negatively affecting protein synthesis and muscle strength [114,115].

Clinically, sarcopenia presents as decreased muscle strength, impaired physical performance, and diminished quality of life, necessitating an integrated approach to diagnosis and management [116].

### 7.3. Emerging Biomarkers and Clinical Implications

In addition to conventional diagnostic tools such as dual-energy X-ray absorptiometry (DEXA) and handgrip strength assessment, recent studies have identified molecular biomarkers that are capable of detecting early sarcopenia in the context of MASLD [117]. Among these, irisin—a myokine derived from the proteolytic cleavage of the FNDC5 precursor—has garnered increasing attention [118]. Reduced circulating levels of irisin have been observed in patients with advanced MASLD and are inversely correlated with hepatic fibrosis severity and transaminase levels [119]. Irisin exerts beneficial effects on mitochondrial biogenesis, lipolysis, and insulin sensitivity, and represents a promising therapeutic target in the management of metabolic–alcoholic sarcopenia [120]. The recognition of sarcopenia as an emerging yet underdiagnosed complication of MASLD carries important implications for risk stratification, clinical surveillance, and the early identification of vulnerable patients. Timely intervention through personalized nutritional programs, targeted physical activity (e.g., resistance training), and selective micronutrient supplementation may disrupt the vicious cycle linking hepatic inflammation, muscle wasting, and clinical deterioration [121,122].

## 8. Moderate Alcohol Consumption: Residual Risk in Metabolic Diseases

### 8.1. Threshold Effects and Recent Evidence

The concept of “moderate alcohol consumption” has historically generated clinical and scientific ambiguity, due to the absence of a universally accepted definition. Widely adopted guidelines define a “moderate” intake as ≤20 g/day for women and ≤30 g/day for men. However, these thresholds are derived from heterogeneous observational studies that are often not adjusted for metabolic comorbidities or genetic polymorphisms affecting ethanol metabolism [123]. A growing body of recent evidence challenges the existence of a “safe” level of alcohol consumption in individuals with metabolic dysfunction [124,125,126]. A prospective study involving over 10,000 individuals with hepatic steatosis demonstrated that even a moderate alcohol intake was significantly associated with an increased risk of advanced hepatic fibrosis, independent of sex, age, and BMI [127]. Similarly, a meta-analysis found that an alcohol intake below 20 g/day was linked to the histological progression of NAFLD/MASLD, particularly in individuals with type 2 diabetes and dyslipidemia [128].

These findings underscore the need for a critical reassessment of the clinical application of the concept of moderation in alcohol consumption.

### 8.2. Mechanisms of Alcohol Vulnerability in MASLD

Patients with MASLD exhibit heightened vulnerability to the effects of alcohol, even at low doses, due to a combination of metabolic, mitochondrial, and immunologic factors [129]. Early induction of CYP2E1, reduced mitochondrial NAD^+^ availability, increased lipid peroxidation, and inefficient detoxification of acetaldehyde collectively contribute to disproportionate oxidative stress and cellular injury [130]. Additionally, the coexistence of intestinal dysbiosis and increased epithelial permeability facilitates bacterial translocation and hepatic inflammation, even in the absence of binge drinking [66,131]. This synergistic interaction between alcohol toxicity and metabolic dysfunction explains the increased risk of progression to MASH, fibrosis, and hepatocellular carcinoma (HCC) observed even in patients with a non-excessive alcohol intake [7,8]. Furthermore, it has been shown that in individuals with MASLD, alcohol consumption impairs the response to dietary and lifestyle interventions, reducing their effectiveness in improving hepatic steatosis and insulin resistance [132].

This metabolic “antagonistic effect” suggests that even low doses of ethanol may meaningfully compromise therapeutic outcomes.

### 8.3. Critical Conclusions on the Clinical Use of the “Moderation” Concept

In light of current evidence, the clinical use of the term “moderate alcohol consumption” appears to be inadequate in the context of MASLD. Safety thresholds are not universal and vary significantly depending on genetic predisposition, body composition, mitochondrial function, and the presence of metabolic comorbidities. Therefore, even in patients with early-stage MASLD or without documented fibrosis, a precautionary approach emphasizing the substantial reduction of or abstinence from alcohol consumption is advisable [133,134]. This strategy is supported by international guidelines, which recommend complete abstinence in individuals with steatohepatitis, advanced fibrosis, or an elevated cardiovascular risk.

In summary, in patients with MASLD, alcohol-related hepatic and systemic risks cannot be neutralized by moderation, nor reliably defined by generalized thresholds.

Personalized risk communication and tailored intervention strategies represent a cornerstone of modern clinical management in metabolic hepatology.

## 9. Nutritional and Clinical Implications

### 9.1. Counseling Strategies and Risk Communication

In the context of MASLD, assessment of alcohol consumption habits should be an integral component of clinical consultation and metabolic risk stratification. The use of validated tools such as AUDIT-C and CAGE enables the early identification of patients with hazardous or harmful drinking behaviors, even in the absence of overt alcohol dependence [135,136]. Effective counseling must be nonjudgmental, empathetic, and patient-centered, fostering empowerment and behavioral change through structured models such as Screening and Brief Intervention (SBI) and the FRAMES framework (Feedback, Responsibility, Advice, Menu, Empathy, Self-efficacy) [137]. It is essential to communicate that no “safe” threshold of alcohol intake exists in the setting of MASLD. The associated risk is modulated by individual factors—such as BMI, diabetes, dyslipidemia, and genetic background—and should be conveyed in a personalized manner.

The engagement of a multidisciplinary team—including a nutritionist, psychologist, and hepatologist—is critical to ensuring ongoing support, measurable goals, and sustainable intervention over time.

Figure 7 outlines a proposed integrated management algorithm for MetALD, combining early screening with targeted lifestyle and nutritional interventions.

### 9.2. Clinical Screening for Fibrosis in Patients with MASLD

Given the high prevalence of hepatic fibrosis among patients with MASLD and metabolic dysfunction, international guidelines recommend a structured screening approach, even in the absence of overt clinical signs [138]. Specifically, in individuals with prediabetes, type 2 diabetes, or metabolic syndrome, the FIB-4 score (based on age, platelet count, AST, and ALT) is indicated as the first-line screening tool [139]. If the FIB-4 score is ≥1.3, transient elastography (FibroScan) is recommended to assess liver stiffness [140,141]. Stiffness values between 8.0 and 12.0 kPa warrant biannual monitoring, while values ≥12.0 kPa necessitate referral to hepatology for further diagnostic evaluation (e.g., liver biopsy) and management of potential complications [142].

Figure 8 illustrates the stepwise algorithm for fibrosis screening in MASLD, integrating the FIB-4 index and transient elastography into a streamlined clinical workflow.

This stepwise pathway enables effective, non-invasive fibrosis assessment, reducing diagnostic delays and optimizing resource allocation in clinical settings with a high prevalence of MASLD.

Key diagnostic biomarkers and non-invasive scoring systems that are applicable to MASLD and MetALD are summarized in Table 6.

### 9.3. Dietary Interventions and Nutritional Strategies: Comparative Effectiveness of Dietary Models in MASLD Management

Dietary intervention is a cornerstone of MASLD management, with the potential to improve steatosis, insulin resistance, and fibrotic progression—even in the absence of approved pharmacologic therapies [143]. Beyond caloric restriction, macronutrient quality and meal timing play critical roles in modulating hepatic inflammation [143]. Among the various dietary models proposed, the Mediterranean diet has the strongest scientific support, owing to its efficacy in reducing steatosis, enhancing insulin sensitivity, and promoting the regression of liver injury [144]. Characterized by high micronutrient density, intake of polyphenols, soluble fiber, and monounsaturated fats (notably from olive oil), the Mediterranean diet exerts anti-inflammatory effects and favorably modulates the gut microbiota [145]. Other dietary approaches—including low-carbohydrate, ketogenic, or high-protein regimens—may offer benefits in specific subgroups (e.g., individuals with obesity or diabetes), but raise concerns regarding long-term adherence, sustainability, and adverse lipid profiles [146,147]. Protein quality (favoring plant-based sources and fish) and the correction of nutritional deficiencies (vitamin D, zinc, magnesium) are essential components, particularly in patients with sarcopenia or ongoing alcohol consumption [148]. To support adherence, interventions can be monitored using validated tools such as the MEDAS score or food frequency questionnaires (FFQs), in addition to digital health applications [149,150].

Integrating nutritional counseling with metabolic treatment and cardiovascular risk management enables a truly personalized and effective approach.

## 10. Conclusions and Future Perspectives

The conceptual shift from NAFLD to MASLD has substantially redefined the diagnostic, therapeutic, and preventive approach to metabolic liver diseases. Within this new paradigm, alcohol consumption—even at levels that are traditionally considered “moderate”—emerges as a significant pathogenic cofactor, capable of exacerbating mitochondrial dysfunction, systemic inflammation, and hepatocellular vulnerability, particularly in individuals with underlying metabolic predisposition. The synergy among mitochondrial dysfunction, impaired intestinal permeability, nutritional depletion, and sarcopenia constitutes an integrated pathophysiological network that accelerates progression to steatohepatitis (MASH), advanced fibrosis, and hepatocellular carcinoma. This systemic model, involving the liver, gut, and skeletal muscle, highlights the need for a more holistic and multidimensional clinical framework—one that incorporates not only biochemical and radiological parameters but also nutritional status, body composition, and behavioral risk factors.

In this context, the systematic adoption of alcohol screening tools (e.g., AUDIT-C), the promotion of micronutrient-rich dietary patterns (e.g., the Mediterranean diet), early monitoring of sarcopenia, and the implementation of structured educational interventions represent evidence-based components of an integrated clinical approach. In patients with MASLD—even at early stages—a precautionary recommendation for alcohol abstinence or significant reduction is warranted, given the lack of universally safe thresholds and the individual variability in susceptibility to ethanol-induced injury.

Nonetheless, critical uncertainties remain that warrant investigation through controlled longitudinal studies. Future research priorities include the following:The validation of specific biomarkers for alcohol-induced liver injury susceptibility in MASLD patients, leveraging multi-omics technologies, quantitative imaging, and immuno-nutritional profiling;The development of non-invasive diagnostic tools tailored to the MetALD phenotype, capable of capturing the specificity of ethanol–metabolism interactions;The evaluation of personalized nutritional and pharmacological interventions, potentially in combination with structured psychological counseling;The integration of liver health into cardiovascular and metabolic prevention programs, recognizing MASLD as a systemic, multi-organ condition.

In summary, the complex interaction between metabolic dysfunction and alcohol consumption represents a paradigmatic challenge for modern medicine. Only an integrated, translational, and multidisciplinary approach—bridging clinical research, personalized care, and informed health policy—can effectively address the growing global burden of metabolic and alcohol-related liver diseases.

## Figures and Tables

**Figure 1 nutrients-17-02229-f001:**
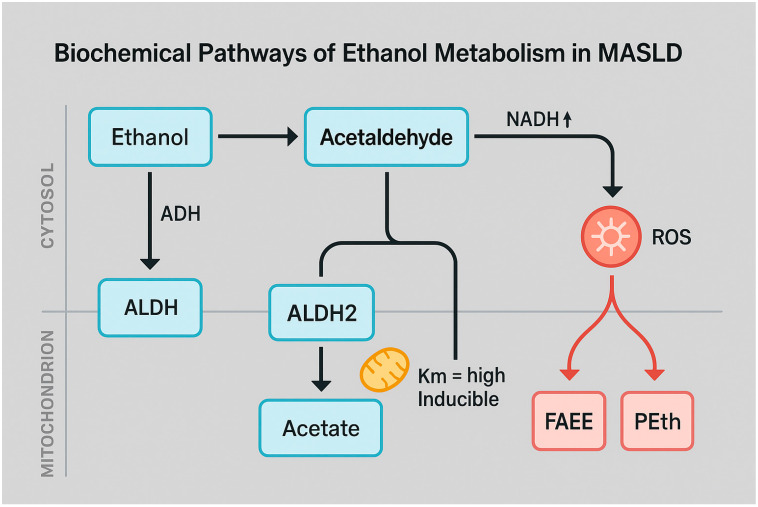
Schematic representation of the principal biochemical pathways involved in hepatic ethanol metabolism and their implications in metabolic dysfunction-associated steatotic liver disease (MASLD). Ethanol is initially oxidized to acetaldehyde by alcohol dehydrogenase (ADH) in the cytosol. Acetaldehyde is then further metabolized in mitochondria by aldehyde dehydrogenase 2 (ALDH2), producing acetate and increasing the intracellular NADH/NAD^+^ ratio. This redox shift impairs mitochondrial β-oxidation and contributes to lipid accumulation. Additionally, acetaldehyde can promote reactive oxygen species (ROS) generation, which exacerbates oxidative stress and activates the formation of non-oxidative ethanol metabolites such as fatty acid ethyl esters (FAEEs) and phosphatidylethanol (PEth), implicated in lipotoxicity. The diagram highlights the mitochondrial localization of ALDH2 and the role of inducible enzymes with high Km. These mechanisms are central to the alcohol-mediated exacerbation of hepatic injury in MASLD. Image created using BioRender (web version accessed April 2025), GraphPad Prism 10.5.0 (May 2025 release), and Microsoft PowerPoint 2021 (Build 16.0.2312.20132).

**Figure 2 nutrients-17-02229-f002:**
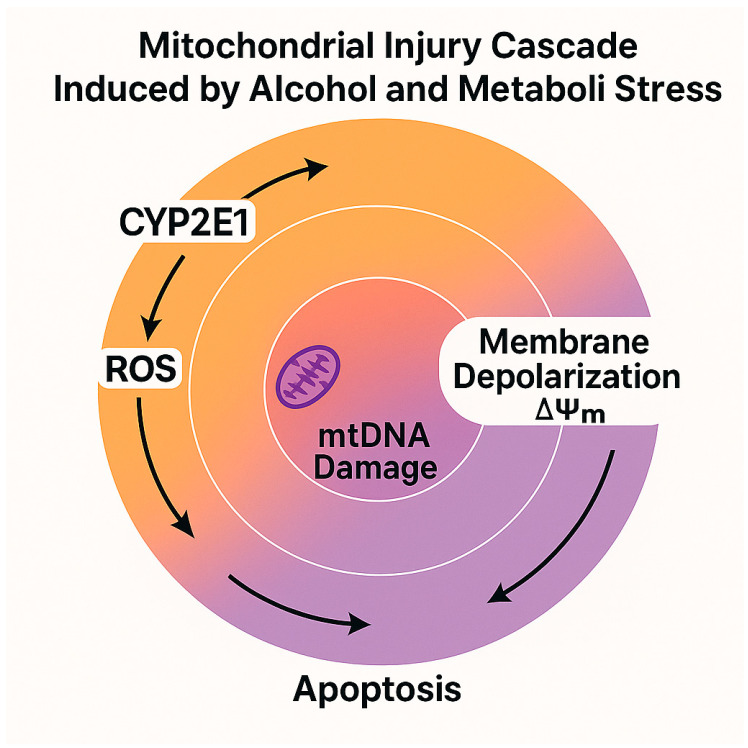
Schematic illustration of the mitochondrial injury cascade induced by alcohol metabolism and metabolic stress in MASLD. The diagram shows a circular progression of mitochondrial damage, beginning with the induction of cytochrome P450 2E1 (CYP2E1), which enhances the generation of reactive oxygen species (ROS). Arrows depict a clockwise sequence where ROS induce mitochondrial DNA (mtDNA) damage, triggering mitochondrial membrane depolarization (ΔΨm) and culminating in apoptosis. The central mitochondrial icon symbolizes the site of mtDNA injury, while concentric gradient rings visually represent the escalating bioenergetic dysfunction. The interplay between ROS, ΔΨm loss, and apoptosis is characteristic of the mitochondrial toxicity observed in metabolic–alcoholic liver disease. Image created using BioRender (web version accessed April 2025), GraphPad Prism 10.5.0 (May 2025 release), and Microsoft PowerPoint 2021 (Build 16.0.2312.20132).

**Figure 3 nutrients-17-02229-f003:**
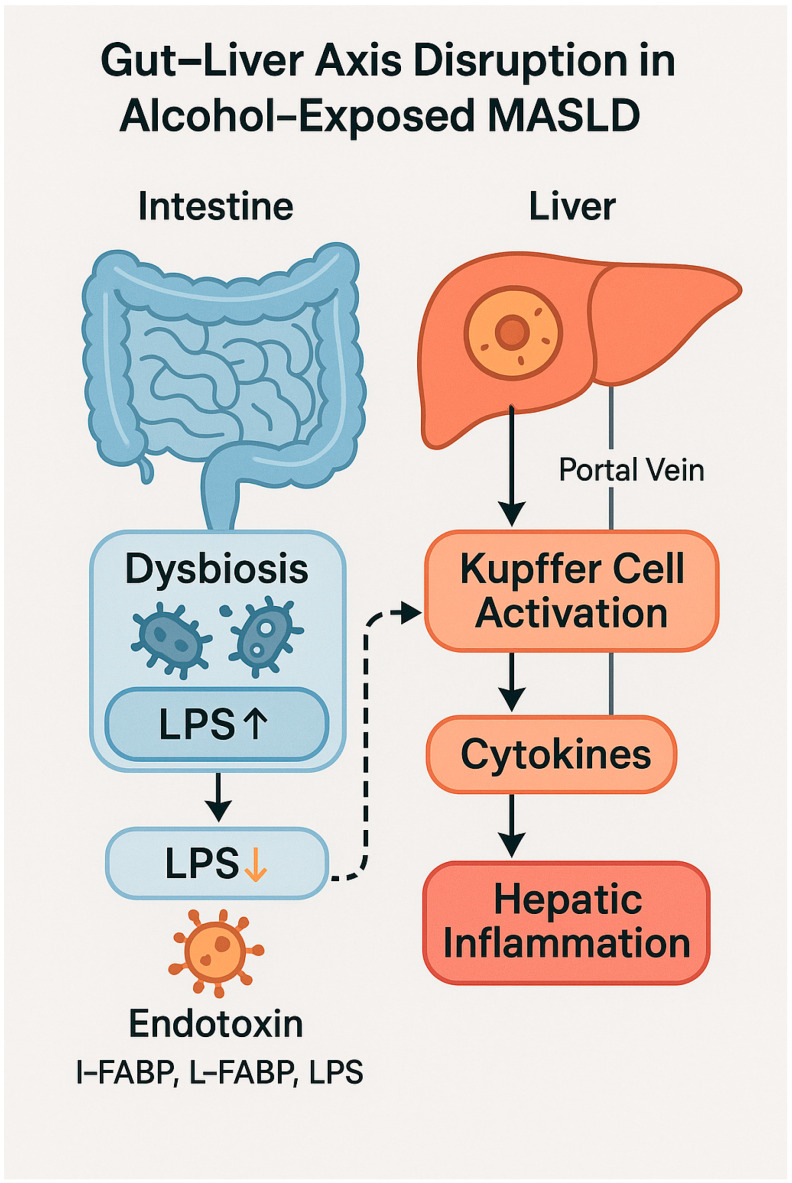
Diagram illustrating the disruption of the gut–liver axis in MASLD with chronic alcohol exposure. On the intestinal side (**left**), alcohol-induced dysbiosis leads to an overproduction of lipopolysaccharide (LPS↑), which increases the luminal endotoxin load. Downward arrows indicate the direction of endotoxin translocation, while the dashed arrow shows the passage of LPS into the portal circulation. Upon reaching the liver (**right**), LPS activates Kupffer cells via Toll-like receptor 4 (TLR4), triggering the release of pro-inflammatory cytokines and promoting hepatic inflammation. Solid arrows between hepatic components represent sequential activation steps. The endotoxin panel at the bottom highlights representative biomarkers of gut barrier dysfunction, including I-FABP, L-FABP, and circulating LPS. Image created using BioRender (web version accessed April 2025), GraphPad Prism 10.5.0 (May 2025 release), and Microsoft PowerPoint 2021 (Build 16.0.2312.20132).

**Figure 4 nutrients-17-02229-f004:**
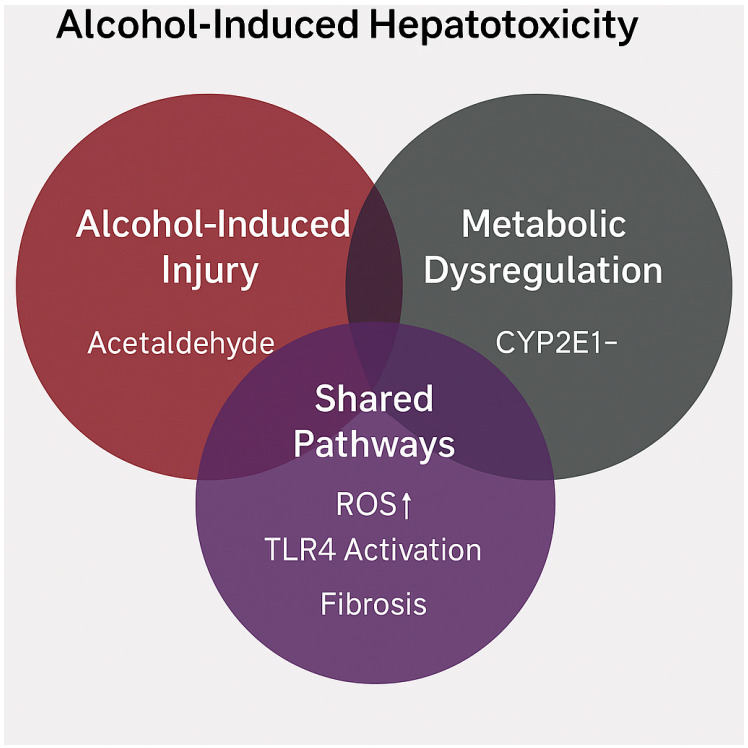
Venn diagram illustrating the overlapping and distinct mechanisms of alcohol-induced hepatotoxicity in the context of MASLD. The left circle highlights alcohol-induced injury, characterized by the accumulation of acetaldehyde and its direct cytotoxic effects. The right circle represents metabolic dysregulation, primarily driven by CYP2E1 overexpression induced by insulin resistance and nutrient excess. The overlapping area identifies shared pathogenic pathways, including increased reactive oxygen species (ROS↑), activation of Toll-like receptor 4 (TLR4), and hepatic fibrogenesis. This visual framework emphasizes the synergistic convergence of toxic and metabolic insults in the MetALD phenotype. Image created using BioRender (web version accessed April 2025), GraphPad Prism 10.5.0 (May 2025 release), and Microsoft PowerPoint 2021 (Build 16.0.2312.20132).

**Figure 5 nutrients-17-02229-f005:**
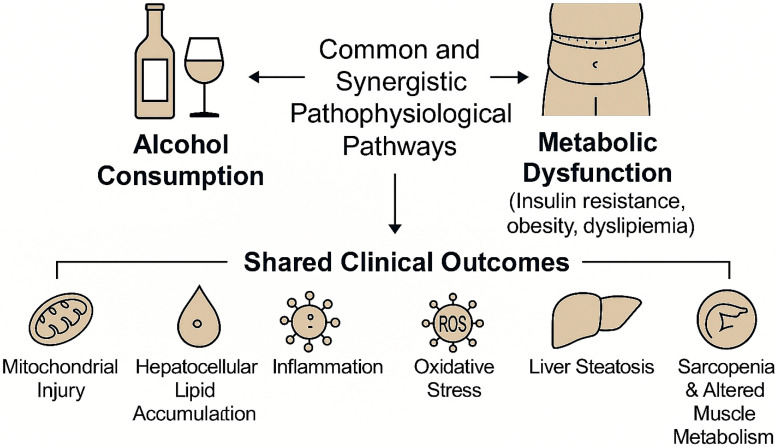
Integrated model of the shared and synergistic pathophysiological pathways linking alcohol consumption and metabolic dysfunction. Both conditions converge on common mechanisms—including mitochondrial injury, hepatocellular lipid accumulation, inflammation, and oxidative stress—culminating in liver steatosis and extrahepatic manifestations such as sarcopenia and impaired muscle metabolism. This framework supports the pathogenic rationale underlying the MetALD phenotype and emphasizes the need for holistic risk stratification. Image created using BioRender (web version accessed April 2025), GraphPad Prism 10.5.0 (May 2025 release), and Microsoft PowerPoint 2021 (Build 16.0.2312.20132).

**Figure 6 nutrients-17-02229-f006:**
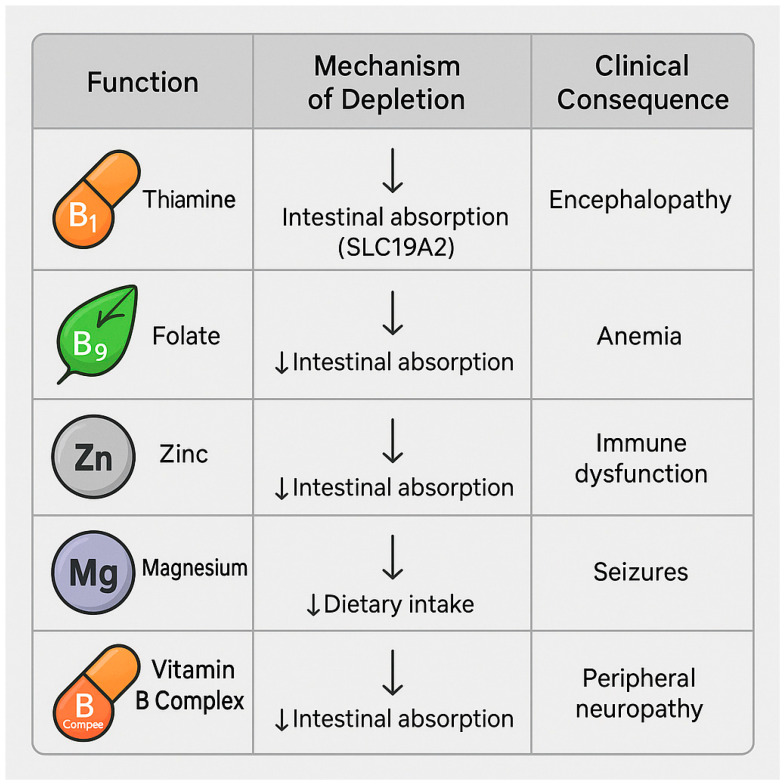
Summary table of micronutrient deficiencies commonly observed in alcohol-exposed MASLD. The diagram lists five key micronutrients (thiamine, folate, zinc, magnesium, and vitamin B complex) alongside their mechanisms of depletion and associated clinical consequences. Downward arrows (↓) indicate reduced intestinal absorption or dietary intake. Thiamine and folate absorption is impaired at the apical membrane of enterocytes, partly due to transporter inhibition (e.g., SLC19A2 for thiamine). Zinc and B-complex vitamins are depleted primarily through decreased intestinal uptake, while magnesium loss is often linked to reduced intake. Corresponding clinical outcomes include hepatic encephalopathy, anemia, immune dysfunction, seizures, and peripheral neuropathy. Image created using BioRender (web version accessed April 2025), GraphPad Prism 10.5.0 (May 2025 release), and Microsoft PowerPoint 2021 (Build 16.0.2312.20132).

**Figure 7 nutrients-17-02229-f007:**
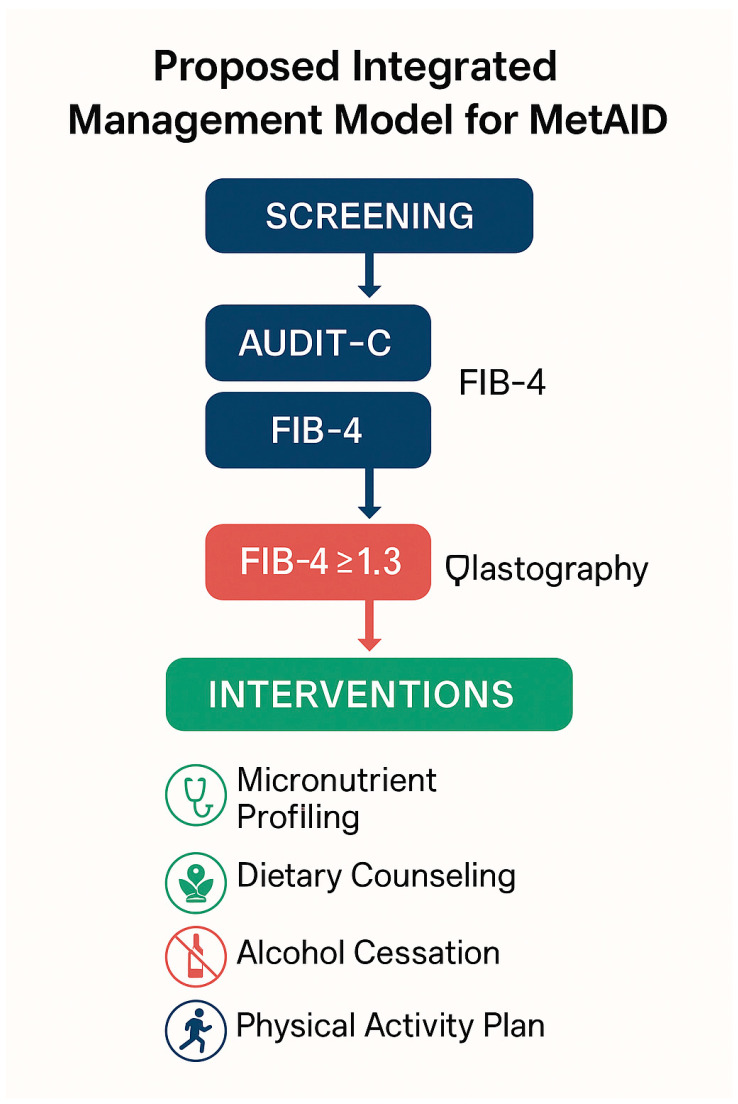
Proposed integrated clinical management model for patients with metabolic and alcohol-related liver disease (MetALD). The pathway includes initial screening using validated tools (e.g., AUDIT-C for alcohol use and FIB-4 for fibrosis risk), followed by transient elastography in patients with FIB-4 ≥ 1.3. Based on stratified risk, a comprehensive set of multidisciplinary interventions is recommended, including micronutrient profiling, personalized dietary counseling, alcohol cessation support, and structured physical activity programs. This model supports the early identification and individualized management of at-risk patients. Image created using BioRender (web version accessed April 2025), GraphPad Prism 10.5.0 (May 2025 release), and Microsoft PowerPoint 2021 (Build 16.0.2312.20132).

**Figure 8 nutrients-17-02229-f008:**
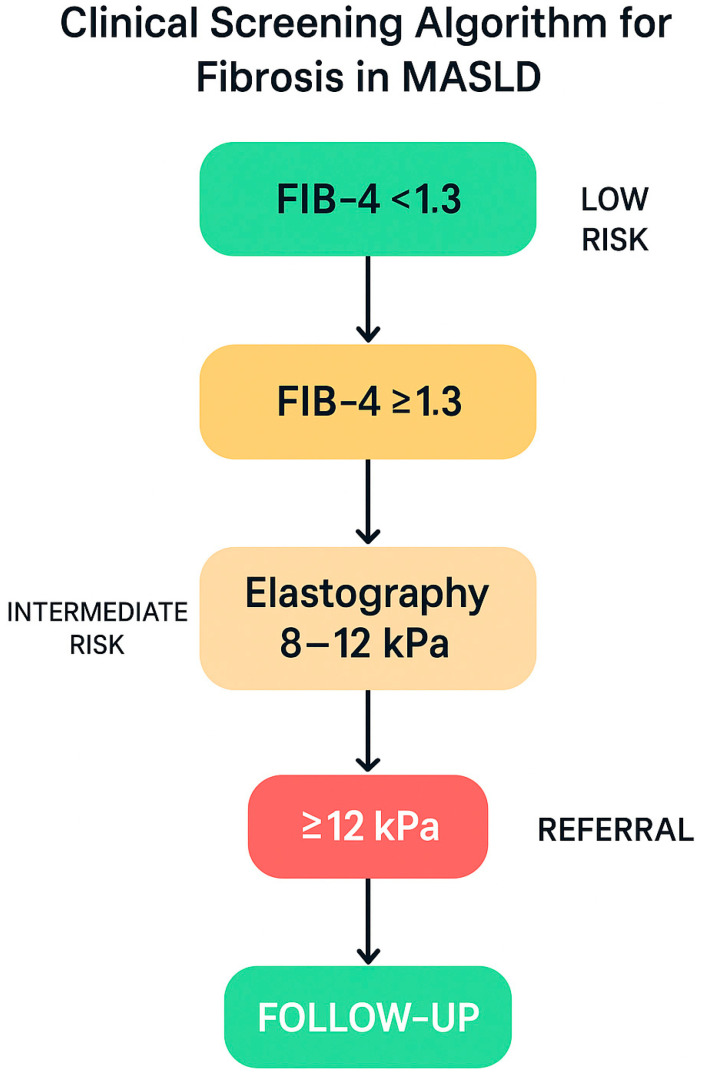
Stepwise clinical screening algorithm for the identification of advanced liver fibrosis in patients with MASLD. Initial stratification is based on the FIB-4 score, with values < 1.3 indicating low risk and annual follow-up. Patients with FIB-4 ≥ 1.3 undergo transient elastography to assess liver stiffness. Intermediate stiffness values (8–12 kPa) warrant biannual monitoring, while values ≥ 12 kPa require specialist referral for further diagnostic work-up and fibrosis management. This non-invasive approach optimizes risk stratification and the early detection of advanced liver disease in metabolic settings. Image created using BioRender (web version accessed April 2025), GraphPad Prism 10.5.0 (May 2025 release), and Microsoft PowerPoint 2021 (Build 16.0.2312.20132).

**Table 1 nutrients-17-02229-t001:** Comparison of MASLD, MetALD, MASH, and ALD across etiology, clinical features, and disease progression.

Feature	MASLD	MetALD	MASH	ALD	References
Main Etiology	Metabolic dysfunction (e.g., obesity, T2DM)	Metabolic dysfunction + moderate alcohol	MASLD with inflammation and hepatocellular injury	Chronic heavy alcohol intake	[2,3,6]
Alcohol Intake	Low to none	Moderate (men < 50 g/day, women < 30 g/day)	Same as MASLD	High (men > 50 g/day, women > 30 g/day)	[6,8,10]
Steatosis	Yes	Yes	Yes	Yes	[2,6]
Inflammation and Ballooning	No (or minimal)	No (or minimal)	Yes	Yes	[3,7,8]
Fibrosis Risk	Variable, increases with comorbidities	Higher than MASLD alone	High (progressive)	High, esp. with sustained alcohol use	[8,9,10]
Progression Speed	Usually slow	Intermediate	Faster than MASLD	Variable; can be rapid	[5,8,9]
Clinical Risk Factors	Obesity, IR, T2DM, dyslipidemia	Same as MASLD + alcohol use	Same as MASLD	Alcohol use disorder, binge drinking	[3,6,9]
Diagnosis Criteria	Imaging or biopsy: steatosis + metabolic criteria	Steatosis + metabolic criteria + moderate alcohol	Histologic: steatohepatitis (inflammation + ballooning)	History of alcohol use + liver injury	[2,6,8]
Treatment Focus	Lifestyle: weight loss, control of metabolic comorbidities	Same as MASLD + reduce/stop alcohol	Intensive lifestyle + monitor inflammation/fibrosis	Abstinence, nutritional support	[8,9,13]
Histology Needed?	Not always	Not always	Yes (for definite diagnosis)	Often used to assess severity	[3,8,9]

MASLD: Metabolic dysfunction-associated steatotic liver disease; MetALD: Metabolic and Alcohol-related Liver Disease; MASH: Metabolic dysfunction-associated steatohepatitis; ALD: Alcohol-associated Liver Disease; T2DM: Type 2 Diabetes Mellitus; IR: Insulin Resistance.

**Table 2 nutrients-17-02229-t002:** Pathogenic mechanisms of alcohol-induced mitochondrial dysfunction in MASLD.

Mechanism	Description	Impact on Liver	References
Excess NADH production	Ethanol oxidation via ADH/ALDH2 generates high NADH levels	Inhibits β-oxidation and gluconeogenesis	[15,17,22]
CYP2E1 overexpression	Induced by alcohol and insulin resistance	Promotes ROS generation, lipid peroxidation, and mitochondrial injury	[18,19,28]
Redox imbalance	Altered NAD^+^/NADH ratio affects metabolic homeostasis	Leads to oxidative stress and ATP production inefficiency	[15,26,31]
Megamitochondria formation	Observed in alcohol-exposed hepatocytes	Reflects impaired mitochondrial dynamics and fusion/fission	[29]
mtGSH depletion	Due to increased ROS and impaired transport	Reduces antioxidant capacity	[30,44]
Mitochondrial respiratory chain damage	Impairs complexes I and III of ETC	Triggers apoptosis via cytochrome *c* release	[28,43]

MASLD: Metabolic dysfunction-associated steatotic liver disease; ADH: Alcohol dehydrogenase; ALDH2: Aldehyde dehydrogenase 2; ROS: Reactive oxygen species; NADH: Nicotinamide adenine dinucleotide (reduced form); mtGSH: Mitochondrial glutathione; ETC: Electron transport chain.

**Table 3 nutrients-17-02229-t003:** Disruption of intestinal barrier and gut–liver axis in alcohol-exposed MASLD.

Mechanism	Description	Hepatic Consequences	References
Dysbiosis	Reduction in SCFA-producing bacteria, increase in pathogens	Promotes endotoxemia and inflammation	[57,58]
Barrier protein loss	Reduced ZO-1 and occludin expression	Increases intestinal permeability	[60]
LPS translocation	Entry of bacterial endotoxin into portal circulation	Activates TLR4 on Kupffer cells, induces cytokines	[60,61]
PAMPs and metabolites	Ethanol-derived and microbial toxins (e.g., DCA, PEth)	Promote hepatic inflammation and FXR/PXR modulation	[64,65]

SCFA: Short-chain fatty acid; ZO-1: Zonula occludens-1; LPS: Lipopolysaccharide; TLR4: Toll-like receptor 4; DCA: Deoxycholic acid; PEth: Phosphatidylethanol; FXR: Farnesoid X receptor; PXR: Pregnane X receptor.

**Table 4 nutrients-17-02229-t004:** Shared and distinct pathophysiological pathways in MASLD and alcoholic liver disease.

Pathway	MASLD	ALD	MetALD Synergy	References
CYP2E1 induction	Moderate, driven by insulin resistance	High, ethanol induced	Synergistic ROS production and mitochondrial injury	[19,73]
Lipotoxicity	Excess nutrients activate SREBP-1c, ChREBP	Acetaldehyde and NADH promote steatosis	Enhanced inflammatory cascades	[24,80,82]
Intestinal permeability	Metabolic inflammation and dysbiosis	Alcohol disrupts tight junctions	Facilitates endotoxemia and fibrosis	[60,66]
Fibrosis progression	Driven by TGF-β, PPAR modulation	Mediated via HSC activation and apoptosis	Accelerated transition to MASH	[28,83,84]

ALD: Alcoholic liver disease; MetALD: Metabolic and Alcohol-related Liver Disease; SREBP-1c: Sterol regulatory element-binding protein-1c; PPAR: peroxisome proliferator–activated receptors; HSC: Hepatic stellate cell; TGF-β: Transforming growth factor-beta.

**Table 5 nutrients-17-02229-t005:** Micronutrient deficiencies in MASLD patients who consume alcohol.

Micronutrient	Function	Consequence of Deficiency	References
Thiamine (B1)	Cofactor in energy metabolism	Lactic acidosis, worsened insulin resistance	[86,87]
Folate	DNA synthesis and methylation	Genomic instability, carcinogenesis risk	[88,90]
Zinc	Enzymatic activity, ammonia detoxification	Encephalopathy, immune dysfunction	[91,92]
Magnesium	Mitochondrial stability, glucose metabolism	Insulin resistance, cramps	[93,94]
B-complex vitamins	Metabolic cofactors	Neuropathy, cognitive deficits	[95,96]

MASLD: Metabolic dysfunction-associated steatotic liver disease.

**Table 6 nutrients-17-02229-t006:** Diagnostic biomarkers and scoring systems in MASLD and MetALD.

Biomarker/Score	Function	Clinical Use	Limitations	References
ALT/AST	Hepatocellular enzymes	Basic liver injury marker	Poor specificity; normal in advanced disease	[5,6,39,139]
Cytokeratin-18 (CK-18)	Apoptosis marker	Marker of steatohepatitis	Variable sensitivity	[39]
FIB-4	Composite index (age, AST, ALT, platelets)	Fibrosis staging	Overlaps with other liver diseases	[39,139,140,141]
ELF score	Fibrosis biomarker panel	Non-invasive fibrosis staging	Limited availability	[39,139,140,141]
Pro-C3	Collagen turnover marker	Advanced fibrosis indicator	Needs standardization	[39,139,140,141]
Hepamet, ADAPT, Agile 3+	Risk scores integrating metabolic and fibrosis parameters	Predict progression, fibrosis	Require validation in MetALD	[39,139,140,141]

ALT: Alanine aminotransferase; AST: Aspartate aminotransferase; FIB-4: Fibrosis-4 Index; ELF: Enhanced Liver Fibrosis; CK-18: Cytokeratin-18.

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
