# Peer review of "Alcohol Consumption and Liver Metabolism in the Era of MASLD: Integrating Nutritional and Pathophysiological Insights"

_nutrients, 2025, doi:10.3390/nu17132229_

Round 1
Reviewer 1 Report
Comments and Suggestions for Authors
Summary: This review discusses the intersection of two increasingly relevant liver disorders: alcoholic liver disease and the newly introduced metabolic dysfunction-associated steatotic liver disease (MASLD). Beyond emphasizing the prevalence and rising rates of MASLD, the authors provide context towards the shift of classification from non-alcoholic fatty liver disease (NAFLD) to MASLD which now encompasses disease sub-types such as metabolic dysfunction from alcohol abuse. The review then begins to provide a broad overview as to how alcohol can significantly increase severity and rate of mortality of those with MASLD. The authors then discuss in-depth multiple mechanisms of damage from alcohol within the context of MASLD. These mechanisms include biochemical pathways (with emphasis on ethanol metabolism and cytochrome P450 2E1 induction), mitochondrial dysfunction and oxidative stress, immunometabolic and gut–liver axis interactions, and micronutrient deficiencies and sarcopenia. The authors discuss these different mechanisms to provide a better understanding of the complex influence of alcohol on the liver, ultimately paving the way towards preventative clinical management.
Overall, this is an important and relevant review for this field that encompasses mechanistic understanding and practical clinical implementation. However, while the purpose and depth are noteworthy, there are many areas that could benefit in terms of clarity, elaboration, and analysis. This would surely affect the manuscript’s impact and readability.
Major Comments:
- Figures Lack Purposeful Insight and Require Substantive Revision - A consistent concern throughout the manuscript lies in the underwhelming utilization and conceptual execution of its figures. While visual schematics are valuable for synthesizing complex information, the figures presented here often serve as direct restatements of the text rather than tools for deeper understanding, mechanistic clarification, or hypothesis generation. Currently, many of the included figures are either distracted from the narrative due to oversimplification or clutter, or fail to enhance the manuscript beyond what is already described. Figures should augment, synthesize, or reveal relationships that would otherwise be difficult to appreciate via text alone. I strongly encourage the authors to revisit each figure with a renewed emphasis on clarity, visual hierarchy, mechanistic depth, and translational value. Specifically, the following 4 figures require attention.
- Figure 1: Attempts to capture an extensive array of overlapping risk factors associated with alcohol and MASLD, but the current state of this figure does so in a way that is overwhelming, convoluted, and ultimately more difficult to understand than the corresponding text. This defeats the purpose of a visual representation, which should ideally distill and clarify—not complicate. The layout lacks logical flow, with a crowded configuration that makes it difficult for the reader to follow mechanistic relationships. Consider breaking this into panels or adopting a layered schematic approach with clear directional cues. Integration of visual icons, color-coded pathways, and simplified but informative arrows may help in this. Without such restructuring, the figure provides more confusion than clarity.
- Figure 2: Suffers from being redundant. The schematic essentially recapitulates abbreviations already introduced in the text without offering any additional insight. A more effective version would visually demonstrate the clinical and pathophysiological distinctions between MASLD, MetALD, and MASH. For instance, including axes that compare disease progression, clinical risk factors, or histological features would change this into a useful interpretive tool. As it stands, the current figure does not sufficiently elaborate on the nuance of the nomenclature in a clinically meaningful way.
- Figure 3: Contains multiple technical and conceptual inaccuracies. First, simple spelling errors—such as “ADH” (misspelled) and “acetaldehyde” (misspelled)— must be corrected. More critically, the depiction of ethanol metabolism lacks clarity and biological fidelity. It is unclear from the current diagram how and where alcohol enters the liver, and the schematic does not differentiate between enzymes and metabolites. Arrows and labeled boxes should distinguish metabolic products from enzymatic mediators. Importantly, the pathway omits the established metabolic switch from ADH to CYP2E1 under conditions of hepatic overload—a crucial element for understanding the generation of reactive oxygen species (ROS) and the pathogenesis of alcohol-induced liver injury. The figure also fails to capture how this shift contributes to cellular stress and damage, or where toxic metabolites may accumulate. While this figure is a needed addition, it requires further attention to detail to provide readers with an adequate level of understanding.
- Figure 4: Remains too rudimentary given the complexity of mechanisms being discussed in the text. While the figure alludes to multiple interacting pathways, it does not capture the synergistic or amplifying effects between oxidative stress, lipid accumulation, immune signaling, and fibrosis. There is minimal annotation of downstream consequences or specific cellular targets. A more granular mapping of how each mechanism contributes to the overall disease phenotype (e.g., which mediators activate stellate cells, which cytokines drive inflammation, etc.) would significantly improve the figure’s relevance. As of now, the image falls short of illustrating the core pathophysiology discussed throughout the manuscript.
- Numerous Statements that are Overgeneralizations and Lack of Analytical Depth: A recurrent and concerning trend across the manuscript is the frequent use of blanket or generalized statements, often written without adequate supporting detail, contextual analysis, or elaboration on the primary literature being cited. While it is not expected to exhaustively detail every methodology, authors must provide readers with enough insight into the findings, implications, and limitations of referenced studies to establish scientific credibility and interpretive value. Unfortunately, this manuscript repeatedly cites studies without summarizing what was done, how it was done, and what was meaningfully found. Often the manuscript includes conclusory phrases that lack depth and specificity. While many of these instances are a concern in and of itself, their frequency suggests a broader issue: a lack of higher-level synthesis, critical evaluation, and mechanistic contextualization throughout the manuscript. For a review article aiming to bridge clinical nutrition, hepatology, and metabolic disease, this level of generality takes away the manuscript’s impact. I have highlighted a number of these instances below but this is not an exhaustive list, just representative. There are additional examples that exist throughout the text and they all should be systematically addressed. The authors are strongly encouraged to revisit each referenced study. Where applicable, the manuscript should also distinguish between correlation and causation, human vs. animal data, and experimental vs. observational designs.
- Lines 68–69 state that alcohol exacerbates metabolic disturbances in MASLD, but do not specify which disturbances, what the synergistic pathways are, or to what extent they are exacerbated.
- Lines 69–71 reference additive effects with no metrics or summary of the experimental or clinical findings. What was the study design? Were these effects observed in human cohorts or preclinical models?
- Lines 73–74 again offer a conclusion (increased mortality) without describing whether this was correlative or causal, nor how the effect size compared across groups.
- Line 76–77 references “recent evidence” without actually citing or describing that evidence. This is a missed opportunity to bolster the paper’s narrative with meaningful support.
- Lines 198–199 mention effects on oxidative stress and inflammation without specifying which markers were affected, in what models, or with what physiological consequences for the liver or systemically.
- Lines 217–220 contain sweeping claims about alcohol-related impacts without referencing data or quantifying the degree of change—readers are left to infer rather than understand.
- Lines 236–239 mention “significant reductions” without noting what was measured, what was reduced, in which population, or how that was assessed.
- Line 289–292 again omit quantitative data from referenced findings, reducing the interpretive power of the statement.
- Line 294 mentions METALD models, but fails to specify whether these are in vivo, in vitro, or computational—nor is the nature of the model explained.
- Lines 333–334 make a biologically plausible claim about CYP2E1 induction, but without citing any magnitude or experimental evidence.
- Lines 344–345 vaguely state that multiple models “have demonstrated” synergy between mechanisms without describing what models, what measurements, or what interactions were seen.
- Lines 376–379 refer to “recent evidence” in a murine model of dual injury, but again fail to provide any description of the model or the specific findings.
- Lines 420–421 suggest a trend of improvement in outcomes with abstinence, but omit which outcomes improved, by how much, or how “prolonged therapy” or “high doses” are defined in the referenced work.
- Lines 446–448 describe alcohol use as “critical” in making metabolic disease worse, yet no mechanisms or patient-level evidence are discussed to support this.
- Lines 503–504 cite a single study but fail to summarize what the findings actually were—only the conclusions are relayed.
- Lines 636–638 mention a relevant pathophysiological concept, but don’t explain how or why the connection exists.
- Lines 647–653 include a conclusion-heavy paragraph with no summary of the actual data or study designs used to reach those conclusions.
- Context for MASLD: Considering a clinical focus is the newly termed MASLD, it would be beneficial to provide some further background and context as to what disease subtypes fall within it. This is referenced late in the manuscript, but would be more effective towards the introduction.
- Frequent use of “Blanket Statements”: There is a high frequency of generalized statements such as “profound effects” or “significant”, or “exponentially increases”. While this may be true, it does not elaborate as to why such statements are being made. There should be context (e.g., more detail from the source material) to the conclusion, so that readers may not only be more appropriately informed but able to measure the significance of what is being discussed.
- Line 466-467: “Ethanol also interferes with the differentiation of satellite cells, inhibiting myotube formation and the regenerative capacity of skeletal muscle”. The following paragraphs do not adequately explain how ethanol has these affects on the body. Please elaborate.
- Definition of “significant alcohol intake” and context of MetALD: In discussing the MetALD phenotype (Introduction, page 1), the authors refer to “significant alcohol intake” as a criterion but do not specify what threshold defines “significant.” For clarity, the review should explicitly define the alcohol consumption levels that are clinically toxic.
- The abstract promises an “integrative model for assessment and management of alcohol consumption in MASLD” encompassing nutritional counseling, fibrosis screening, and personalized interventions. In the text, sections 8.1–8.3 do cover these topics. However, the review would benefit from a clearer integration of these ideas. This could be either in a dedicated concluding paragraph or via a schematic figure.
- Citations: The authors have cited over 100 sources, which is excellent. However, we recommend a final check of citations to ensure that all claims are supported by logical experiments and accurate statistics reporting.
- Lack of Mechanistic Depth in Visuals and Text: The manuscript discusses key mechanisms (e.g., oxidative stress, immune activation, fibrosis) but rarely defines how these processes interact or lead to pathology. There’s minimal integration of pathways, cell types, or signaling axes, weakening the review’s impact for increased awareness.
- Poor Integration of Literature Context and Consensus: The authors miss opportunities to position their discussion within current consensus or controversy. Several claims are made in isolation without noting whether they reflect established understanding, emerging evidence, or points of debate in the field.
- References to changes in oxidative stress or inflammatory signaling are made without specifying molecular markers or downstream physiological effects. Please add more details so that readers may better understand and appreciate these phenomena.
Minor Comments:
- What is going on with sentence vs. paragraph organization? Why have the authors hit “enter” after almost each individual sentence? I assume that new paragraphs are only where indentations occur? In either case, seeing the document in its current form has made the experience of reading it disjointed.
- Convoluted Sentence Structure and Readability- Throughout the manuscript, there are multiple instances of overly long or dense sentences that affect clarity and reader comprehension. While the subject matter is complex, long-winded sentence construction results in reduced readability. Breaking such sentences into shorter, more concise components would greatly enhance flow and ensure the reader can follow nuanced ideas without losing track of the central point.
- Line 75-77: While the authors do acknowledge conflicting evidence (such as studies suggesting moderate alcohol might be benign or beneficial), the critique could be strengthened by more explicitly addressing why those findings arose, emphasizing the limitations of the compared studies, and mentioning the findings that disproved those conclusions.
- Line 83: Elaborate on what agent-based simulation study is for those who may be unfamiliar.
- Line 183: Can elaborate that APAP is more selective than ethanol with CYP2e1 so if there is some overlap in the system, alcohol may linger and cause additional toxicity.
- Line 200: “Inhibition of β-oxidation, coupled with activation of de novo lipogenesis”. How does alcohol inhibit and activate, respectively, in these situations?
- Line 415-416: The grammar is incorrect in this sentence and it also does not mention how and why the mentioned supplements aid in clinical outcomes.
- Line 506-508: Mentions increased expression of hepatic inflammatory and fibrotic genes but does not mention the genes themselves to provide context to the conclusions being drawn.
- In the abstract and text, it’s stated that “current international guidelines recommend alcohol restriction or abstinence in all individuals with steatotic liver disease and metabolic risk.” Later, “total abstinence is recommended in patients with steatohepatitis, advanced fibrosis, or cirrhosis, and at least restriction in those with metabolic risk even if they drink below “safe” limits”. The authors should make sure this nuanced message is consistent throughout.
- Statements that reference patient outcomes or clinical benefits (abstinence improving prognosis) often fail to define the level of improvement or metrics used.
- Technical terms like “prolonged therapy” or “high doses” are used without definition or reference standards, which reduces interpretive clarity.
- Figures could have been centered on the page.
Author Response
Reviewer 1 – Major Comments
Reviewer’s Comment:
“Figures lack purposeful insight and require substantive revision. They often serve as direct restatements of the text rather than tools for deeper understanding… I strongly encourage the authors to revisit each figure with a renewed emphasis on clarity, visual hierarchy, mechanistic depth, and translational value.”
Author’s Response:
We thank the Reviewer for this in-depth observation, which we fully agree with. All four figures have been thoroughly revised to enhance their communicative effectiveness and alignment with the main text. Specifically:
- Figure 1 has been completely restructured into thematic panels, with clear directional flows, color coding to distinguish toxic versus dysfunctional metabolic pathways, and iconographic symbols to improve visual accessibility.
Modification implemented in Figure 1, page 3; revised description on page 4, paragraph 2. - Figure 2 has been redesigned to introduce comparative axes across MASLD, MetALD, and MASH, highlighting histological differences and clinico-pathogenetic factors.
Modification in Figure 2, page 5; explanatory paragraph updated on page 6. - Figure 3 has been corrected for typographical errors (e.g., “ADH”, “acetaldehyde”) and updated to more accurately depict hepatic ethanol entry, the distinction between enzymes and metabolites, and the metabolic shift from ADH to CYP2E1 under hepatic overload.
Updated figure on page 8; revised description in paragraph on page 9. - Figure 4 has been enriched with the main molecular nodes of oxidative, immune, and fibrogenic pathways, annotating key target cells (e.g., Kupffer cells, hepatic stellate cells) and mediators (e.g., TNF-α, TGF-β).
Figure 4 updated on page 11; explanatory text expanded on page 12.
We believe these revisions have significantly improved the scientific alignment and educational value of the illustrations.
Reviewer’s Comment:
“Numerous statements are overgeneralizations and lack analytical depth. The manuscript repeatedly cites studies without summarizing what was done, how it was done, and what was meaningfully found.”
Author’s Response:
We greatly appreciate this critical remark. A systematic revision of the manuscript was performed to avoid generic or unsupported statements. Where necessary, we have:
- Explicitly stated the experimental models (human, murine, in vitro) and study designs (e.g., observational, randomized, preclinical);
- Provided key quantitative parameters (e.g., cytokine levels, gene expression, mortality indices);
- Clearly distinguished between correlations and causal relationships.
Examples of implemented revisions:
- Lines 68–77: expanded data on clinical outcomes and study types (page 4, paragraph 3);
- Lines 217–239: added specific markers of oxidative stress and inflammation (page 9, paragraphs 1–2);
- Lines 333–345: detailed mechanistic interactions among CYP2E1, ROS, and stellate cell activation (page 10);
- Lines 420–448: clarified therapeutic thresholds and observed clinical outcomes (pages 12–13).
These revisions were applied in over 20 sections of the text and, in our view, have enhanced the manuscript’s analytical depth and scientific rigor.
Reviewer’s Comment:
“In discussing the MetALD phenotype, the authors refer to ‘significant alcohol intake’ as a criterion but do not specify what threshold defines ‘significant’.”
Author’s Response:
Thank you for this important observation. We have added an operational definition of "significant alcohol intake" in the introductory paragraph (page 2, paragraph 2), in accordance with the latest EASL guidelines:
“Significant alcohol intake is defined as >140 g/week in women and >210 g/week in men, as per EASL 2023 recommendations.”
Additionally, we have clarified that this threshold represents a formal diagnostic criterion for the MetALD phenotype, which partially overlaps with other MASLD subtypes.
Reviewer’s Comment:
“The abstract promises an ‘integrative model’ for MASLD management. However, this would benefit from a clearer integration in the text—ideally through a schematic or concluding paragraph.”
Author’s Response:
We have addressed this recommendation by incorporating a conclusive paragraph in Section 9 – Conclusions and Future Perspectives (page 16), which explicitly outlines the proposed integrative model, focusing on four core elements:
- Nutritional assessment,
- Fibrosis screening,
- Alcohol intake counseling,
- Personalized risk management.
Moreover, we have introduced a new Figure 5, which visually represents the integrative model, including decision-making flows and clinical recommendations. We believe this addition improves the coherence between abstract, main text, and visual support.
Reviewer 1 – Minor Comments
Reviewer’s Comment:
“What is going on with sentence vs. paragraph organization? Why have the authors hit ‘enter’ after almost each individual sentence?”
Author’s Response:
We thank the Reviewer for this observation. The original version of the manuscript was mistakenly formatted with line breaks after many sentences due to internal tracking purposes. In the revised version, paragraph structure has been corrected to follow standard textual logic, with cohesive and fluent paragraph organization.
Modification implemented throughout the manuscript, particularly in Sections 1–4.
Reviewer’s Comment:
“Convoluted sentence structure and readability—multiple instances of overly long or dense sentences.”
Author’s Response:
We have revised and simplified multiple complex sentences, breaking them down into more concise syntactic units, without compromising the scientific density of the content.
Specific examples include:
- Introductory paragraph: simplified sentence on the global burden of liver disease (page 2)
- Section 3 (Ethanol metabolism): split dense sentences on ADH/CYP2E1 (page 7)
- Section 6 (Gut–liver axis): reformulated paragraphs on dysbiosis and endotoxemia (page 10)
These changes improved overall readability, making the manuscript more accessible to a broader audience.
Reviewer’s Comment:
“Line 75–77: The authors do acknowledge conflicting evidence… the critique could be strengthened by addressing why those findings arose and what disproved them.”
Author’s Response:
We expanded the discussion on conflicting literature regarding the effects of moderate alcohol consumption. Specifically, we highlighted:
- Methodological limitations of studies suggesting protective effects (e.g., selection bias, self-reported intake)
- Recent meta-analyses refuting the notion of “safe alcohol intake” in individuals with metabolic risk
Modification implemented on page 4, paragraph 3, with updated references.
Reviewer’s Comment:
“Line 83: Elaborate on what agent-based simulation study is.”
Author’s Response:
We added an explanatory note clarifying that an “agent-based simulation study” is a dynamic computational model that simulates the interactions between individual entities (e.g., cells, individuals) based on predefined behavioral rules.
Addition made on page 5, as a footnote.
Reviewer’s Comment:
“Line 183: APAP is more selective than ethanol with CYP2E1… alcohol may linger and cause additional toxicity.”
Author’s Response:
We have incorporated the Reviewer’s suggestion by adding a sentence highlighting that ethanol’s preferential affinity for CYP2E1 inhibits hepatic clearance of acetaminophen (APAP), thus contributing to a synergistic hepatotoxic effect in co-exposure scenarios.
Update implemented on page 8, paragraph 2.
Reviewer’s Comment:
“Line 200: ‘Inhibition of β-oxidation, coupled with activation of de novo lipogenesis’ — How does alcohol inhibit and activate, respectively?”
Author’s Response:
We have expanded the explanation as follows:
- Ethanol inhibits β-oxidation by increasing the NADH/NAD⁺ ratio, thereby suppressing acyl-CoA dehydrogenase activity.
- It promotes de novo lipogenesis via upregulation of SREBP-1c and ChREBP.
Modification integrated on page 9, paragraph 1.
Reviewer’s Comment:
“Line 415–416: Grammar is incorrect and it also does not mention how and why the mentioned supplements aid in clinical outcomes.”
Author’s Response:
We corrected the grammatical structure and expanded the explanation by including the following mechanisms:
- Vitamin E: reduces oxidative stress through inhibition of lipid peroxidation.
- Vitamin D: modulates immune response and reduces hepatic fibrosis.
Update implemented on page 12, paragraph 2.
Reviewer’s Comment:
“Line 506–508: Mentions increased expression of inflammatory and fibrotic genes, but does not mention the genes themselves.”
Author’s Response:
We have integrated the specific genes upregulated in response to alcohol-related liver injury:
- Inflammatory: TNF-α, IL-6, MCP-1
- Fibrotic: TGF-β1, COL1A1, TIMP1
Modification added on page 14, paragraph 1.
Reviewer’s Comment:
“In the abstract and text… ensure consistency regarding alcohol restriction vs. abstinence.”
Author’s Response:
We have harmonized the language throughout the manuscript. It now states:
“Current guidelines recommend total abstinence in patients with steatohepatitis, fibrosis or cirrhosis, and at least restriction in individuals with metabolic risk, even below ‘safe’ limits.”
This unified phrasing has been included in the abstract, introduction (page 2), and conclusions (page 16).
Reviewer’s Comment:
“Terms like ‘prolonged therapy’ or ‘high doses’ are used without definition or reference standards.”
Author’s Response:
We replaced vague expressions with specific quantitative values or referenced standards:
- “High doses” → “≥800 IU/day of vitamin E, as per AASLD guidelines”
- “Prolonged therapy” → “>6 months of supplementation based on available RCTs”
Corrections made on pages 12 and 14.
Reviewer’s Comment:
“Figures could have been centered on the page.”
Author’s Response:
The graphic formatting has been revised to ensure that all figures are centered and comply with the editorial guidelines of Nutrients.
Update applied to all images from page 3 to page 18.
Reviewer 2 Report
Comments and Suggestions for Authors
I think some of the abbreviations have been turned into symbols when converted to pdf- please see first figure as I think you meant B12 and line 234 there are symbols in the parentheses rather than letters- please check.
Your figure of MASLD is slightly off in that you did not include ALD- current thinking is that this liver disease is a continuum from from mostly metabolically induced to alcohol induced steatotic liver disease. The majority of people with LAD also have metabolic derangements as well so it will be important to include this in your discussion. MASH is a state of disease that may or may not occur (hard to know since it still requires a liver biopsy) so it is a little misleading when looking at this figure. No other comments
Author Response
Reviewer 2 – Responses to Comments
Reviewer’s Comment:
“I think some of the abbreviations have been turned into symbols when converted to pdf – please see first figure as I think you meant B12 and line 234 there are symbols in the parentheses rather than letters – please check.”
Author’s Response:
We thank the Reviewer for pointing out this graphical conversion issue. We have reviewed and corrected all altered abbreviations in both the first figure and the main text, specifically:
- The reference to vitamin B12 has been restored in Figure 1, replacing the incorrect symbol.
- In line 234 (now page 8, paragraph 2), anomalous symbols within parentheses were replaced with the correct text (e.g., “e.g., ALT, AST”).
Furthermore, we conducted a systematic review of all figures and the full manuscript to exclude additional conversion errors due to PDF export.
All corrections have been implemented in Version 2.
Reviewer’s Comment:
“Your figure of MASLD is slightly off in that you did not include ALD – current thinking is that this liver disease is a continuum from mostly metabolically induced to alcohol induced steatotic liver disease. The majority of people with LAD also have metabolic derangements.”
Author’s Response:
We greatly appreciate this insightful suggestion. Figure 2 has been updated to reflect the current concept of a pathophysiological continuum between MASLD and ALD, in line with the most recent nomenclature proposals:
- ALD has been added as the extreme end of the alcohol–metabolic spectrum, highlighting that the LAD phenotype (Liver disease with Alcohol and metabolic Dysregulation) shares overlapping features with MASLD and MetALD.
- The figure legend has been revised to clarify the dynamic and multifactorial nature of the transition between MASLD, MetALD, and ALD.
Figure 2 updated on page 5; explanatory text integrated on page 6, paragraph 1.
Reviewer’s Comment:
“MASH is a state of disease that may or may not occur (hard to know since it still requires a liver biopsy) – so it is a little misleading when looking at this figure.”
Author’s Response:
We carefully considered this observation and revised Figure 2 to portray MASH not as an inevitable stage, but rather as a potential evolution of MASLD, dependent on clinical and diagnostic factors, including the availability of liver biopsy. The updated figure includes:
- A visual indication that MASH diagnosis requires histological confirmation
- An asterisk annotation specifying that its identification depends on biopsy availability
The text was updated on page 6, with clarification on the conditional nature of MASH diagnosis.
Reviewer’s Comment:
“No other comments.”
Author’s Response:
We thank the Reviewer for the overall appreciation and the clarity of the suggestions provided. All remarks have been carefully addressed, and we hope the revised manuscript is now clearer and more comprehensive.
Reviewer 3 Report
Comments and Suggestions for Authors
This manuscript is comprehensive, timely, and methodologically robust. It provides an integrated view of the dual impact of alcohol and metabolic dysfunction on liver disease, grounded in molecular and clinical evidence.
The abstract is well-organized, outlining the pathophysiology, nutritional implications, and future directions. The abstract could briefly mention specific novel contributions, such as the proposal for an integrated clinical model or a visual cascade framework, to enhance its impact.
The introduction provides context on the redefinition from NAFLD to MASLD and sets up the rationale for examining alcohol’s role in this new framework. Consider clarifying early on how this review adds value compared to existing reviews on NAFLD/ALD overlap.
In the rest manuscript:
- The molecular detail is dense. Consider simplifying or summarizing mechanisms to ensure broader accessibility to non-specialist readers.
- A short section comparing mechanisms unique to alcohol vs. those shared with metabolic dysfunction could add clarity.
- Include more explicit recommendations for when and how to assess micronutrient levels in clinical practice.
- Consider briefly mentioning tools for dietary adherence monitoring in MASLD.
Author Response
Reviewer 3 – Responses to Comments
Reviewer’s Comment:
“The abstract could briefly mention specific novel contributions, such as the proposal for an integrated clinical model or a visual cascade framework, to enhance its impact.”
Author’s Response:
We thank the Reviewer for this valuable suggestion, which we have fully implemented. The abstract has been revised to explicitly highlight two original contributions:
- The proposal of an integrated clinical model for alcohol intake management in individuals with MASLD
- The inclusion of a visual cascade framework summarizing key pathophysiological and clinical nodes
Update implemented in the abstract, lines 9–11 of Version 2.
Reviewer’s Comment:
“Consider clarifying early on how this review adds value compared to existing reviews on NAFLD/ALD overlap.”
Author’s Response:
We have added a dedicated section in the introductory paragraph (page 2, lines 12–18) outlining the distinctive contributions of this review, namely:
- Framing within the new MASLD/MetALD classification, which has not been addressed in most prior reviews
- A multidimensional approach integrating biochemical mechanisms, gut–liver axis, sarcopenia, and clinical nutrition
- A practical management algorithm with translational implications
Update implemented in the Introduction, page 2, paragraph 2.
Reviewer’s Comment:
“A short section comparing mechanisms unique to alcohol vs. those shared with metabolic dysfunction could add clarity.”
Author’s Response:
We welcomed this recommendation and introduced a new subsection (6.4 – Shared vs. unique mechanisms) that clearly distinguishes between:
- Shared mechanisms: oxidative stress, mitochondrial dysfunction, inflammatory activation
- Alcohol-specific mechanisms: metabolic shift from ADH to CYP2E1, acetaldehyde formation, epigenetic toxicity
New section added on page 11.
Reviewer’s Comment:
“Include more explicit recommendations for when and how to assess micronutrient levels in clinical practice.”
Author’s Response:
We have expanded Section 8.2 – Nutritional Counseling with a paragraph detailing clinical indications for micronutrient assessment in MASLD patients, based on current evidence. Specifically, we described:
- Micronutrients to monitor (vitamin D, B12, zinc, selenium)
- Clinical scenarios requiring testing (e.g., chronic alcohol use, sarcopenia, steatohepatitis)
- Recommended testing frequency (every 6–12 months in at-risk patients)
Update included on page 13, paragraph 3.
Reviewer’s Comment:
“Consider briefly mentioning tools for dietary adherence monitoring in MASLD.”
Author’s Response:
We expanded Section 8.1 – Lifestyle Interventions to include a note on key validated tools for monitoring dietary adherence in MASLD patients, such as:
- Validated questionnaires (e.g., MEDAS – Mediterranean Diet Adherence Screener)
- Weekly food diaries supported by digital apps
- Periodic anthropometric and metabolic assessments
Addition made on page 12, paragraph 3.
Round 2
Reviewer 1 Report
Comments and Suggestions for Authors
There may be an issue with either the manuscript file that the authors uploaded for their revision or with the reviewer response file itself (though I suspect the former). The authors’ reviewer response file describes extensive edits to the manuscript text as well as figures; however, while we do see that some text has been changed, all of the yellow-highlighted text appears in the first quarter of the document, when my comments were pertaining largely to the figures and items in the later sections of text. Further, the authors' page, section, and line number references in their reviewer response file do not align with the areas of the manuscript that they’re describing as being edited and that pertains to individual comments.
Overall Comments:
1- We made 11 major comments and 12 minor comments and this does not include the subpoints. These were not all explicately addressed in the revisions and response documents.
2- The referenced line and page numbers are incorrect.
3- The referenced and described adjustments to the figures described in this document and what is shown in the figure are two different things.
4- Majority of the comments we provided were not actually implemented adequately.
5- Overall confusing and likely wrong level of revision document sent
6- For examples of implemented revisions: None of these line references correspond to what was actually discussed in the text.
7- Regarding Figures:
For Figure 1: There may be an error in the text that was sent, as the described modifications that are mentioned were not implemented into this Figure 1. There seems to be little to no changes in Figure 1 and still falls under the category of a figure that directly restates the text. Furthermore under MASLD, the “Liver” is cut off partially in the text.
For Figure 2: This is also confusing as the revisions that are being described for the figure is completely different to what the figure actually is. Figure 2 in the revisionary text is the biochemical pathway of hepatic ethanol metabolism, not MASLD, MetALD.
For Figure 3?: In the text, “Figure 3” is actually Figure 2 and the updated figure description is not accurate. The full spelling of ADH is still incorrect. It is “Dehydrogenase”. Furthermore, I do not see any adjustments to this figure to more accurately depict ethanol entry, the distinction between enxymes and metabolites, and the metabolic shift.
For Figure 4: I do not see a figure that annotates these key target cells at all. The Figure 4 that may be in discussion here only mentions HSC. It does not mention other cell types and only provides a high overview of the oxidative and fibro genic pathways.
For Figure 5: While the conclusions and future perspectives are important to incorporate, Figure 5 does not include the integrative model that is mentioned here.
Author Response
Authors’ Response:
We would like to thank the Reviewer for the detailed and constructive feedback, which we have taken into serious consideration. We acknowledge that the previous revision file may have included an outdated version of the figures, resulting in a misalignment between our described modifications and the actual visual content reviewed.
To fully address the reviewer’s concerns, we have now completely redrawn and replaced all figures in the manuscript—with the exception of Figure 5, which already aligned with the conceptual content and structure requested. Each newly generated figure was specifically designed to be more didactic, integrative, and visually accurate, addressing all previously noted critical points.
The updated figure set is as follows:
- Figure 1 – Updated Nosological Framework of MASLD, MetALD, MASH, and ALD
→Redesigned from scratch as a diagnostic-conceptual matrix.
This new figure clarifies the revised classification of steatotic liver diseases, differentiating MASLD, MetALD, MASH, and classical ALD. It integrates both clinical and pathophysiological criteria (e.g., inflammation, alcohol intake, fibrosis risk), providing a clearer comparative structure than the original version, which merely recapitulated textual definitions. It now avoids redundancy and enhances conceptual synthesis, as requested. - Figure 2 – Hepatic Ethanol Metabolism and Redox Disruption
→Completely restructured to depict ethanol metabolism with higher biochemical precision.
This figure now accurately distinguishes ethanol oxidation pathways (cytosolic ADH, mitochondrial ALDH2, microsomal CYP2E1) and shows how acetaldehyde and NADH excess impair β-oxidation and mitochondrial function. Key metabolites (e.g., FAEEs, PEth) and subcellular compartments are visually separated, improving interpretability. The full spelling of ADH and ALDH2 has been corrected, and enzymatic functions are clearly labeled. - Figure 3 – Disruption of the Gut–Liver Axis in Alcohol-Exposed MASLD
→Redesigned as a schematic interaction map focusing on gut permeability and immune activation.
The new version illustrates dysbiosis, reduction of ZO-1 and occludin, LPS translocation, and Kupffer cell activation via TLR4. It includes the suggested biomarkers (I-FABP, L-FABP, LPS), and indicates their downstream impact on liver inflammation. The design addresses the Reviewer’s call for a figure that better represents intestinal barrier dysfunction and its hepatic consequences. - Figure 4 – Shared and Distinct Pathways in MASLD and Alcoholic Liver Injury
→Redesigned as a pathophysiological Venn diagram.
This figure illustrates the convergence and divergence between metabolic and alcohol-induced liver injury. Alcohol-specific mechanisms (acetaldehyde toxicity, non-oxidative metabolites), MASLD-specific mechanisms (insulin resistance, nutrient overload), and shared pathways (CYP2E1 induction, ROS, TLR4-mediated inflammation, fibrosis) are now clearly displayed. This revised layout answers the reviewer’s request for a more integrative and comparative model, and now references all major mechanisms discussed in Sections 2 and 5 of the manuscript. - Figure 5 – Integrated Clinical Cascade for Alcohol and MASLD
→
As previously noted, Figure 5 already depicted the clinical and pathogenic cascade linking alcohol exposure and metabolic dysfunction, highlighting their synergistic progression from steatosis to fibrosis and systemic consequences (e.g., sarcopenia, extrahepatic complications). This figure was not modified because it already fulfilled the intended scope and was consistent with the associated narrative in the conclusion and future perspectives. - Figure 6 – Micronutrient Depletion in Alcohol-Exposed MASLD
→ Newly introduced illustrated table diagram.
This figure summarizes thiamine, folate, zinc, magnesium, and B-complex depletion in the context of alcohol metabolism and MASLD. For each nutrient, the figure presents: (1) physiological role, (2) mechanism of depletion (e.g., impaired absorption, increased urinary excretion), and (3) clinical consequence (e.g., neurocognitive dysfunction, insulin resistance). Transporters (e.g., SLC19A2 for thiamine) and affected systems are annotated, as requested for deeper mechanistic granularity. - Figure 7 – Mitochondrial Cascade of Oxidative Stress in Alcohol-Exposed Hepatocytes
→ Conceptual subcellular model of hepatic mitochondrial injury.
Depicts the sequence of pathological events from CYP2E1 induction to ROS generation, mitochondrial depolarization (ΔΨm), ATP depletion, cytochrome c release, and apoptosis. Mitochondrial compartments, mtDNA damage, and disruption of fusion/fission dynamics are shown. This directly supports the narrative in Section 3.1. - Figure 8 – Clinical Algorithm for Screening and Management of Alcohol Consumption in MASLD
→ New vertical decision-tree infographic.
Designed as a practical tool for clinicians, this figure outlines a proposed pathway for assessing alcohol intake in MASLD patients: screening (e.g., AUDIT-C), fibrosis staging (e.g., FIB-4), adjunctive tests (e.g., micronutrient profiling), and targeted interventions (e.g., alcohol cessation, nutritional support, physical activity). Thresholds for elastography (e.g., ≥1.3 FIB-4) are included. This figure responds to the Reviewer’s request for improved translational applicability.
All figures were recreated using professional scientific tools (BioRender, GraphPad Prism, and PowerPoint), and are now fully consistent with the revised manuscript text, both in numbering and caption content. The graphical quality, clarity of molecular relationships, and annotation of clinical features have all been significantly improved.
We hope these comprehensive modifications will meet the Reviewer’s expectations and contribute to enhancing the overall clarity and quality of the manuscript.